# Conditional Diffusion Sampling

**Francisco M. Castro-Macías** [1] [*] **Pablo Morales-Álvarez** [1] **Saifuddin Syed** [2] **Daniel Hernández-Lobato** [3]
**Rafael Molina** [1] **José Miguel Hernández-Lobato** [4]

## Abstract

Sampling from unnormalized multimodal distributions with limited density evaluations remains a fundamental challenge in machine learning and natural sciences. Successful approaches construct a bridge between a tractable reference and the target distribution. Parallel Tempering (PT) serves as the gold standard, while recent diffusion-based approaches offer a continuous alternative at the cost of neural training. In this work, we introduce Conditional Diffusion Sampling (CDS), a framework that combines these two paradigms. To this end, we derive Conditional Interpolants, a class of stochastic processes whose transport dynamics are governed by an exact, closed-form stochastic differential equation (SDE), requiring no neural approximation. Although these dynamics require sampling from a non-trivial initialization distribution, we show both theoretically and empirically that the cost of this initialization diminishes for sufficiently short diffusion times. CDS leverages this by a two-stage procedure: (1) PT is used to efficiently sample the initial distribution, and then (2) samples are transported via the transport SDE. This combination couples the robust global exploration of PT with efficient local transport. Experiments suggest that CDS has the potential to achieve a superior trade-off between sample quality and density evaluation cost compared to state-of-the-art samplers.

## 1. Introduction

We consider the problem of drawing independent samples from a target probability distribution $\nu$ defined on a state

---

*Work done while visiting the University of Cambridge. [1]University of Granada [2]University of British Columbia [3]Universidad Autónoma de Madrid (EPS and CIAFF) [4]University of Cambridge. Correspondence to: Francisco M. Castro-Macías <fcastro@ugr.es>.

*Proceedings of the $43^{rd}$ International Conference on Machine Learning*, Seoul, South Korea. PMLR 306, 2026. Copyright 2026 by the author(s).

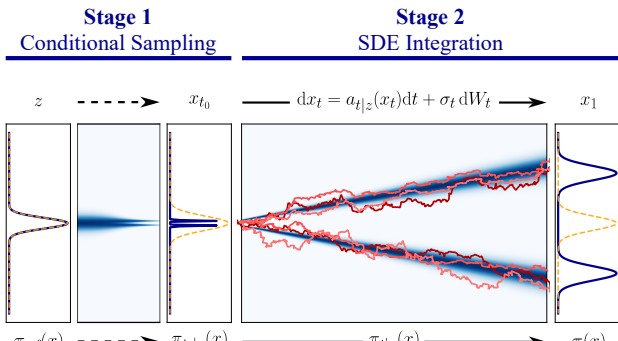

**Stage 1**
Conditional Sampling

**Stage 2**
SDE Integration

$z \quad \cdots\cdots\blacktriangleright \quad x_{t_0} \quad \underline{\quad} \mathrm{d}x_t = a_{t|z}(x_t)\mathrm{d}t + \sigma_t\,\mathrm{d}W_t \quad\longrightarrow\quad x_1$

$\pi_{\mathrm{ref}}(x) \quad \cdots\cdots\blacktriangleright \quad \pi_{t_0|z}(x) \quad \underline{\qquad\qquad} \pi_{t|z}(x) \underline{\qquad\qquad}\longrightarrow \pi(x)$

*Figure 1.* **Overview of CDS.** In the first stage, Parallel Tempering (PT) transforms initial samples $z$ from the reference $\pi_{\mathrm{ref}}$ (orange) into samples from the initialization distribution $\pi_{t_0|z}$ (blue). In the second stage, these samples are transported to the target distribution $\pi$ (blue) by integrating the closed-form SDE in Eq. 15.

space $\mathcal{X} \subset \mathbb{R}^D$. We assume that $\nu$ admits a positive density $\pi\colon \mathcal{X} \to (0, +\infty)$ known only up to a normalization constant. This means that we have access only to the unnormalized density $\tilde{\pi}\colon \mathcal{X} \to (0, +\infty)$ and

$$\pi(x) = Z^{-1}\tilde{\pi}(x), \quad Z = \int_{\mathcal{X}} \tilde{\pi}(x)\,\mathrm{d}x. \qquad (1)$$

This classical problem arises across machine learning (Izmailov et al., 2021) and the natural sciences (Kroese et al., 2014). In these applications, evaluating $\tilde{\pi}$ is often computationally expensive, e.g., due to molecular force-field calculations (Noé et al., 2019) or large neural networks (Izmailov et al., 2021). Consequently, it is crucial to design samplers that achieve high sample quality with as few density evaluations as possible.

A common strategy is to construct a bridge between a tractable reference distribution and the target. A prominent example is Parallel Tempering (PT, Geyer (1991); Hukushima & Nemoto (1996)), which defines a sequence of intermediate distributions and runs multiple Markov chains in parallel, exchanging states to propagate information toward the target. Modern non-reversible variants of PT dominate their reversible counterparts (Syed et al., 2022), but can be slow to converge if the reference shares meagre overlap with the target (Surjanovic et al., 2024).

More recently, generative modeling has popularized

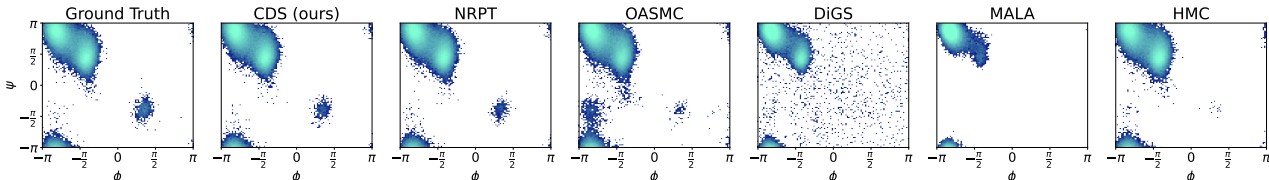

*Figure 2.* **Ramachandran histograms of Alanine Dipeptide (ALDP) in vacuum at** $T = 300$K**.** All methods utilize a fixed budget of $2 \cdot 10^5$ density evaluations. Only the proposed CDS and NRPT successfully capture all modes under this limited budget.

diffusion-based approaches for bridging distributions (Song et al., 2021), in which samples are transported continuously via a diffusion or flow. This perspective has been unified under the theory of Stochastic Interpolants (Albergo et al., 2025), and extended to unnormalized targets via Neural Diffusion Samplers (Akhound-Sadegh et al., 2024; Nusken et al., 2024; Albergo & Vanden-Eijnden, 2025; Akhound-Sadegh et al., 2025; Zhang et al., 2025). However, these methods typically require either training on pre-existing data or learning transport maps through iterative optimization, both of which are costly in terms of target density evaluations (He et al., 2025b). Nevertheless, once trained, they can generate samples without evaluating the target density. As such, they operate in a complementary, amortized regime.

In this work, we introduce Conditional Interpolants, a special class of interpolants that follow a conditional, rather than marginal, probability path. Their associated transport dynamics admit a closed-form, learning-free expression, with drift depending only on the target score and the interpolant map, both of which are tractable. This exactness imposes a constraint: the dynamics cannot be initialized at $t = 0$ and instead require samples from the probability path at some $t_0 > 0$.

We show that this requirement is, in fact, advantageous. As $t_0 \to 0$, the probability path concentrates toward a Dirac measure centered at a reference sample, making the sampling process of the initialization distribution highly efficient. Building on this insight, we propose Conditional Diffusion Sampling (CDS), a two-stage method that (i) efficiently samples from the intermediate distribution using PT for near-zero $t_0$, and (ii) transports these samples to the target via the closed-form conditional diffusion. An overview of CDS is shown in Fig. 1.

Our contributions are: (i) the derivation of Conditional Interpolants, a general class of stochastic interpolants with exact, closed-form dynamics; (ii) a theoretical analysis of the initialization distribution, showing vanishing transport cost as diffusion time decreases, alongside empirical evidence of efficient PT sampling; (iii) the introduction of CDS, combining global exploration via PT with efficient conditional diffusion; and (iv) extensive evaluation across 8 target distributions and 4 tasks, demonstrating a superior trade-off between sample quality and density evaluation costs.

## 2. Background

### 2.1. Markov Chain Monte Carlo

Markov Chain Monte Carlo (MCMC, (Meyn & Tweedie, 2012)) methods aim to sample from an unnormalized target distribution $\nu$ by constructing a Markov chain whose invariant distribution is $\nu$. This is typically achieved by designing a transition kernel that leaves $\nu$ invariant, such as Metropolis–Hastings (MH, Metropolis et al. (1953); Hastings (1970)), MALA (Roberts & Tweedie, 1996), or Hamiltonian Monte Carlo (HMC, Duane et al. (1987)). We provide a brief introduction to these kernels in Subsec. D.2.

**Annealing-based Methods.** In high-dimensional and multimodal settings, standard MCMC kernels often suffer from poor mixing, becoming trapped in individual modes and producing highly correlated samples (Meyn & Tweedie, 2012). This motivates annealing-based strategies, which introduce a sequence of annealing distributions $\nu_\beta$ interpolating between a tractable reference $\nu_{\text{ref}}$ (e.g., a Gaussian or tempered target) at $\beta = 0$ and the target $\nu$ at $\beta = 1$. A common choice is the geometric path, with densities $\pi_\beta$ given by

$$\pi_\beta(x) \propto \pi_{\text{ref}}(x)^{1-\beta} \, \pi(x)^\beta, \tag{2}$$

where $\pi_{\text{ref}}$ is the density of the reference distribution. Given a schedule $0 = \beta_0 < \cdots < \beta_N = 1$, samples are progressively transported from $\nu_{\text{ref}}$ to $\nu$ through these intermediate distributions. Importantly, performance is highly sensitive to the schedule and degrades when the overlap between the reference and the target is poor (Syed et al., 2021; 2024).

**Parallel Tempering (PT).** PT is an annealing-based method that runs several chains in parallel, one per distribution in the annealing schedule. It alternates between local MCMC updates and swap moves between adjacent chains. Low-$\beta$ chains explore smoother landscapes, and swap moves propagate this exploration toward the target, improving global mixing. We provide more details in App. A.

### 2.2. Diffusions and Interpolants

**Diffusions.** A stochastic process $\{x_t\} = \{x_t : 0 \leq t \leq 1\}$ is an (Itô) diffusion process if it is the solution to an stochastic differential equation (SDE) of the form

$$\mathrm{d}x_t = a_t(x_t)\mathrm{d}t + \sigma_t \mathrm{d}W_t, \tag{3}$$

where $x_t \in \mathbb{R}^D$, $(t, x) \mapsto a_t(x) \in \mathbb{R}^D$ is the drift vector field, $t \mapsto \sigma_t \in (0, +\infty)$ is the diffusion coefficient, and $W_t$ is a $D$-dimensional Wiener process. In the limit where the diffusion coefficient vanishes ($\sigma_t \to 0$), the dynamics reduce to a deterministic flow governed by an Ordinary Differential Equation (ODE), with marginals satisfying the continuity equation. We discuss this connection in App. B.

If the marginal $x_t$ at time $t$ admits a time-dependent density $x \mapsto p_t(x)$, then $p_t$ satisfies the Fokker-Planck-Kolmogorov (FPK) equation (Särkkä & Solin, 2019):

$$\frac{\partial}{\partial t} p_t = -\operatorname{div}(p_t a_t) + \frac{\sigma_t^2}{2} \Delta p_t. \tag{4}$$

Conversely, if $p_t$ solves the FPK equation and the coefficients $a_t$, $\sigma_t$ satisfy standard regularity and growth conditions, then there exists a stochastic process solving the SDE in Eq. 3 in the weak sense whose time marginals are given by $p_t$ (Särkkä & Solin, 2019). Under appropriate choice of drift term $a_t$, one may enforce $p_0 = \pi_{\text{ref}}$ (a fixed tractable reference distribution), and $p_1 = \pi$ (the target distribution) (Song et al., 2021). In this case, the SDE in Eq. 3 defines a continuous probabilistic bridge between $\pi_{\text{ref}}$ and $\pi$.

**Stochastic Interpolants.** Albergo et al. (2025) provides a unifying framework for flows and diffusions, known as Stochastic Interpolants. Instead of defining the process via an SDE or ODE directly, one specifies the process explicitly as an interpolation between samples:

$$x_t = F_t(z, x), \quad z \sim \nu_{\text{ref}}, \quad x \sim \nu, \tag{5}$$

where $\nu$ is the target distribution and $\nu_{\text{ref}}$ is a tractable reference distribution, and $F_t$ is a differentiable map satisfying the boundary conditions $F_0(z, x) = z$ and $F_1(z, x) = x$.

*Example* 2.1 (Linear Interpolant). We consider the linear interpolant as a running example throughout this work. For $\mathcal{X} = \mathbb{R}^D$, it is defined as:

$$x_t = F_t(z, x) = (1 - t)z + tx, \tag{6}$$

with $z \sim \nu_{\text{ref}}$, and $x \sim \nu$. This is the canonical choice in flow matching frameworks (Lipman et al., 2023).

The resulting marginals define a probability path from $\nu_{\text{ref}}$ to $\nu$. Crucially, the framework guarantees the existence of a drift field $a_t$ such that the marginals of $x_t$ satisfy the FPK equation (or a velocity field $u_t$ satisfying the continuity equation in the deterministic limit). This duality allows one to characterize the interpolant $x_t$ equivalently as the solution to a diffusion SDE or a flow ODE.

In practice, the vector fields governing these dynamics are intractable and are approximated using neural networks trained on samples from $\nu$. Since we assume no prior access

to such samples, this approach is unsuitable. Yet, the interpolant perspective motivates our construction. In the next section, we introduce Conditional Interpolants, a class of interpolants admitting closed-form dynamics, and leverage them to design Conditional Diffusion Sampling (CDS).

## 3. Conditional Diffusion Sampling (CDS)

**Overview.** First, we provide a high-level overview of Conditional Diffusion Sampling (CDS), our framework for sampling from unnormalized multimodal distributions.

The central idea of CDS is to reduce a difficult global sampling problem into two manageable stages. We condition on a reference point $z \sim \nu_{\text{ref}}$ and construct a *conditional path of distributions* $\{\nu_{t|z}\}_{t \in (0,1]}$ that gradually evolves from a point mass at $z$ to the full target distribution $\nu$.

The path is defined via a *Conditional Interpolant* map $F_{t|z}$ (defined in Subsec. 3.1), which allows us to derive closed-form transport dynamics that need a careful initialization. As illustrated in Fig. 1, our sampling procedure leverages this in two stages:

**Stage 1: Conditional Sampling (Subsec. 3.3).** We initialize the process at a small time $t_0 > 0$ by sampling $x_{t_0} \sim \nu_{t_0|z}$. As illustrated in Fig. 3, $\nu_{t_0|z}$ highly concentrates around $z$, ensuring substantial overlap with the reference distribution $\nu_{\text{ref}}$, and making global exploration highly efficient. In practice, using Parallel Tempering (PT) to sample $\nu_{t_0|z}$ yields improved swap efficiency and mode exploration, see Subsec. 5.1.

**Stage 2: SDE Integration (Subsec. 3.2).** We transport the approximate samples from $\nu_{t_0|z}$ to the target $\nu$ at $t = 1$. Because the interpolant dynamics admit a closed-form SDE, we can transport and refine these samples without any approximation. Crucially, the SDE provides continuous refinement, correcting samples along the trajectory, see Subsec. 5.2.

The remainder of this section details each component: Subsec. 3.1 introduces Conditional Interpolants, Subsec. 3.2 derives their transport dynamics (Stage 2), Subsec. 3.3 analyzes the initialization (Stage 1), and Subsec. 3.4 presents the full algorithm. Notably, Stage 2 is presented before Stage 1, as the derivation of the transport dynamics clarify the need for careful initialization.

### 3.1. Conditional Interpolants

Let $\nu_{\text{ref}}$ denote a tractable reference distribution (with density $\pi_{\text{ref}}$) from which sampling is straightforward. Following Albergo et al. (2025), we consider stochastic interpolants as given by Eq. 5. Rather than focusing on the marginal distribution of $x_t$, we consider the conditional distribution of $x_t$ given a reference variable $z \sim \nu_{\text{ref}}$.

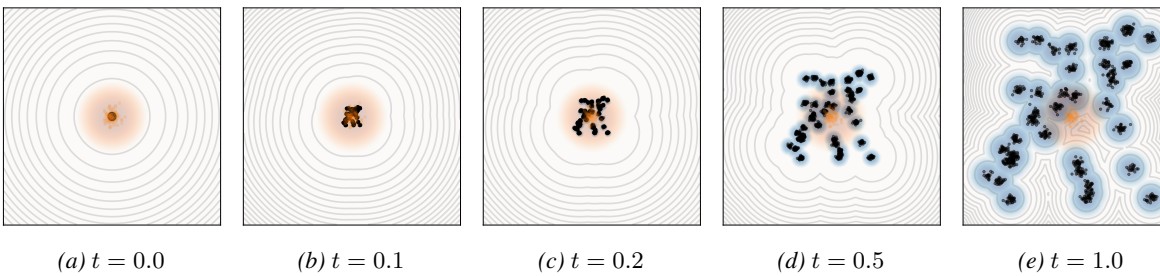

*(a) $t = 0.0$*   *(b) $t = 0.1$*   *(c) $t = 0.2$*   *(d) $t = 0.5$*   *(e) $t = 1.0$*

*Figure 3.* **Density evolution and exact samples from $\pi_{t|z}$.** This plot illustrates the linear interpolant for a fixed $z$ sampled from $\mathcal{N}(0, I)$. As $t \to 0$, the target distribution (blue) increasingly concentrates inside the reference (orange).

Fix $z \sim \nu_{\text{ref}}$ and define the conditional interpolation map $F_{t|z}(\cdot) = F_t(z, \cdot)$, so that

$$x_t = F_{t|z}(x), \quad x \sim \nu. \tag{7}$$

**Definition.** A *Conditional Interpolant* is the family of random variables defined by Eq. 7, where for each $z \sim \nu_{\text{ref}}$ and $t \in (0, 1]$, the map $F_{t|z} : \mathcal{X} \to \mathcal{X}$ is a diffeomorphism. This induces a conditional distribution $\nu_{t|z}$ as the pushforward of $\nu$ through $F_{t|z}$.

**Conditional Density.** Let $\nu_{t|z}$ denote the distribution of the random variable $x_t$ conditioned on $z$. Its density, denoted $\pi_{t|z}$, is determined by the change of variables formula:

$$\pi_{t|z}(x) = \left| \det \mathrm{J} F_{t|z}\big(F_{t|z}^{-1}(x)\big) \right|^{-1} \pi\big(F_{t|z}^{-1}(x)\big), \tag{8}$$

where $\mathrm{J}F_{t|z}$ denotes the Jacobian matrix of the map $F_{t|z}$.

*Example* 3.1. For the linear interpolant defined in Eq. 6, the conditional density takes the closed form:

$$\pi_{t|z}(x) = t^{-D} \pi \left( \frac{x - (1 - t)z}{t} \right). \tag{9}$$

Under standard regularity assumptions, as $t \to 0$, the measure $\nu_{t|z}$ concentrates mass around $z$. Formally,

$$\lim_{t \to 0} W_1(\delta_z, \nu_{t|z}) = 0, \tag{10}$$

where $\delta_z$ is a Dirac delta centered at $z$, and $W_1(\cdot, \cdot)$ denotes the Wasserstein-1 distance; see Lemma D.1 for a proof. Intuitively, $\nu_{t|z}$ interpolates between a Dirac delta $\delta_z$ at $t = 0$ and the target distribution $\nu$ at $t = 1$, see Fig. 3. Indeed, $\nu_{t|z}$ corresponds to the displacement interpolant between $\nu$ and $\delta_z$ (McCann, 1997).

*Remark* 3.2. We can recover the marginal of $x_t$ from this conditional distribution as

$$\pi_t(x) = \int_{\mathcal{X}} \pi_{t|z}(x)\, \pi_{\text{ref}}(z)\, \mathrm{d}z. \tag{11}$$

## 3.2. Transport Dynamics

We characterize the dynamics associated with the proposed family of interpolants. First, we define the conditional velocity field $u_{t|z}$ as the instantaneous time-evolution of the interpolant map:

$$u_{t|z}(x) = \frac{\partial F_{t|z}}{\partial t} \left( F_{t|z}^{-1}(x) \right). \tag{12}$$

This vector field describes the deterministic trajectory of the interpolant defined in Eq. 7.

*Example* 3.3. For the linear interpolant introduced in Eq. 6, the velocity field takes the simple form:

$$u_{t|z}(x) = \frac{x - z}{t}. \tag{13}$$

The velocity field $u_{t|z}$ governs the deterministic transport of probability mass, but does not use any information about the target distribution; we elaborate about this in App. B. To account for this, we can introduce stochasticity while preserving the marginal distributions. The following proposition establishes the link between the interpolant's density and a specific FPK equation.

**Proposition 3.4.** *For any $t \mapsto \sigma_t \in (0, \infty)$, the conditional density satisfies the following FPK equation:*

$$\frac{\partial}{\partial t} \pi_{t|z} = -\operatorname{div}\left( \pi_{t|z} a_{t|z} \right) + \frac{\sigma_t^2}{2} \Delta \pi_{t|z}, \tag{14}$$

*where $a_{t|z} = u_{t|z} + \frac{\sigma_t^2}{2} \nabla \log \pi_{t|z}$.*

*Proof.* See Proposition D.4. □

Motivated by this, we define the corresponding Conditional Diffusion via the following SDE:

$$\mathrm{d}x_t = \left( u_{t|z}(x_t) + \frac{\sigma_t^2}{2} \nabla \log \pi_{t|z}(x_t) \right) \mathrm{d}t + \sigma_t\, \mathrm{d}W_t. \tag{15}$$

Here, the noise schedule $\sigma_t$ controls the path stochasticity without altering the time-dependent density $\pi_{t|z}$ (Song et al., 2021). Its impact is analyzed in App. H.

A key advantage over standard score-based generative modeling is that the score function, $\nabla \log \pi_{t|z}$, does not need to be learned. Instead, it is calculated directly from the target density $\pi$ via the change of variables formula and can be evaluated efficiently (see Lemma D.3).

*Example* 3.5. For the linear interpolant, the score term in the SDE simplifies to:

$$\nabla \log \pi_{t|z}(x) = t^{-1} \nabla \log \pi \left( \frac{x - (1-t)z}{t} \right). \quad (16)$$

The dynamics in Eq. 15 balance the deterministic transport $u_{t|z}$ with a stochastic correction guided by the score. The noise term allows the trajectory to explore the state space, while the score term continuously corrects the path towards high-density regions of the target $\pi$.

**Singularities and Initialization.** Numerical integration of Eq. 15 requires careful handling of the boundary at $t = 0$. Since the interpolant $F_{t|z}$ is non-invertible at $t = 0$, the velocity field $u_{t|z}$ exhibits a singularity at $t = 0$, causing trajectories to diverge if integrated directly from $t = 0$. This behavior mirrors the singularities observed in diffusion models and flow matching, where vanishing noise variance leads to unbounded scores or vector fields (Song et al., 2021; Lipman et al., 2023).

To circumvent this, we initialize the process at a near-zero time $t_0 > 0$. Crucially, as shown in App. H, a deterministic initialization $x_{t_0} = z$ yields poor performance because the diffusion cannot sufficiently expand from a point mass to cover the support of $\nu_{t_0|z}$. Therefore, we initialize by sampling $x_{t_0} \sim \nu_{t_0|z}$. We discuss efficient sampling strategies for this distribution in the following subsection.

### 3.3. Sampling the Initial State

We aim to sample from $\nu_{t_0|z}$ for near-zero $t_0 > 0$. As $t \to 0$, the measure $\nu_{t|z}$ concentrates around the anchor point $z$, converging to the Dirac measure $\delta_z$. Accordingly, samples from this regime are expected to lie close to $z$.

To formalize this behavior, we analyze how a Markov kernel transforms when its invariant distribution is modified from $\nu$ to $\nu_{t|z}$. The following result relates convergence under the transformed kernel to the Lipschitz properties of the interpolant. See Subsec. D.2 for background on Markov kernels.

**Proposition 3.6.** *Let $K$ be any Markov kernel invariant with respect to $\nu$. We define the rescaled Markov kernel $K_{t|z}$ for any $x \in \mathcal{X}$ and measurable set $A \subset \mathcal{X}$ as:*

$$K_{t|z}(x, A) = K \left( F_{t|z}^{-1}(x), F_{t|z}^{-1}(A) \right). \quad (17)$$

*Then, the kernel $K_{t|z}$ is invariant with respect to $\nu_{t|z}$. Furthermore, let $L_t$ be the Lipschitz constant of the interpolant $F_{t|z}$. If we initialize the chain at the reference point $z$, the Wasserstein-1 distance to the target satisfies:*

$$W_1 \left( K_{t|z}^n \left( \delta_z \right), \nu_{t|z} \right) \leq L_t \, W_1 \left( K^n \left( \delta_{x_0} \right), \nu \right), \quad (18)$$

*where $x_0 = F_{t|z}^{-1}(z)$.*

*Proof.* See Proposition D.5. $\square$

Equation 18 implies that whenever $L_t \leq 1$, the sampling error – measured by the Wasserstein-1 distance – is strictly lower when targeting $\nu_{t|z}$ than when targeting the original distribution $\nu$. Moreover, if $\lim_{t \to 0} L_t = 0$, the error vanishes as $t$ decreases. While this property does not hold for arbitrary bijections, it is satisfied by standard interpolants used in generative modeling, such as linear and trigonometric paths (Albergo et al., 2025). Geometrically, contracting the space reduces the transport cost between the initialization and the target $\nu_{t|z}$ proportionally.

*Example* 3.7. In the linear interpolant case, the bound in Eq. 18 becomes an equality, implying that the error decays exactly linearly with $t$:

$$W_1 \left( K_{t|z}^n \left( \delta_z \right), \nu_{t|z} \right) = t \, W_1 \left( K^n \left( \delta_z \right), \nu \right). \quad (19)$$

See Corollary D.6 for a proof.

Thus, for any fixed number of MCMC steps $n$, samples become arbitrarily close to $\nu_{t|z}$ in the optimal transport sense as $t \to 0$.

However, a small transport cost to $\nu_{t_0|z}$ does not necessarily imply faster mixing. For non-linear interpolants, geometric distortions induced by the condition number of $F_{t|z}$ may hinder exploration. Even in the linear case, $\nu_{t_0|z}$ retains the multimodal structure of $\nu$, differing only by a global contraction. Consequently, standard MCMC methods may still mix poorly. To mitigate this issue, we employ PT to sample from $\nu_{t_0|z}$, as described in the next section.

### 3.4. Algorithmic Details

The procedure of CDS is summarized in Alg. 1. It samples from the target distribution $\nu$ in two stages: (i) sampling from the conditional distribution $\nu_{t_0|z}$, and (ii) integrating the conditional SDE in Eq. 15.

**Computational Budget Allocation.** Given a fixed computational budget, performance depends on how computation is split across the two stages. We control this trade-off via two hyperparameters: the number of steps in Stage 1 ($K$)

---

**Algorithm 1** Conditional Diffusion Sampling (CDS)

---

**Require:** Number of PT steps $K$, time schedule $\{0 < t_0 < \cdots < t_N = 1\}$, noise schedule $\{\sigma_0, \ldots, \sigma_{N-1}\}$, number of corrector steps $M$.

---

**STAGE 1: Conditional Sampling**

1: $z \sim \nu_{\text{ref}}$
2: // Run PT to sample from $\nu_{t_0|z}$.
3: // Starting from $z$, $K$ steps.
4: $x_0 \leftarrow \text{PTSampler}\left(z, \nu_{t_0|z}, K\right)$

**STAGE 2: SDE Integration**

5: **for** $n = 0$ **to** $N - 1$ **do**
6:     $\xi \sim \mathcal{N}(0, I)$
7:     $\Delta t_n \leftarrow t_{n+1} - t_n$
8:     $a_n \leftarrow u_{t_n|z}(x_n) + \frac{\sigma_n^2}{2} \nabla \log \pi_{t_n|z}(x_n)$
9:     $y \leftarrow x_n + \Delta t_n a_n + \sigma_n \sqrt{\Delta t_n} \xi$
10:    // Run a corrector to sample from $\nu_{t_{n+1}|z}$.
11:    // Starting from $y$, $M$ steps.
12:    $x_{n+1} \leftarrow \text{Corrector}\left(y, \nu_{t_{n+1}|z}, M\right)$
13: **end for**
14: **return** $x_N$

---

and in Stage 2 ($N$). We find that sufficient effort must be devoted to Stage 1 to ensure enough global exploration, while retaining enough integration steps to accurately transport the samples, see Subsec. H.5.

**Choice of Sampler in Stage 1.** Stage 1 aims to sample from $\nu_{t_0|z}$ for $t_0 > 0$. While any sampler could be used, $\nu_{t_0|z}$ remains multimodal, making local MCMC methods prone to poor mixing. Annealing-based methods are better suited, as they progressively bridge the reference and target distributions, and benefit from the fact that $\nu_{t_0|z}$ approaches the reference as $t_0 \to 0$. In practice, we find that PT outperforms other choices, see Subsec. H.2.

**Numerical Integration.** We use the Euler–Maruyama scheme to integrate the SDE, though any suitable solver could be employed. The diffusion variance $\sigma_t$ controls the stochasticity of the process, and can be specified as a noise schedule. Following diffusion models (Song et al., 2021), integration can be enhanced with an $M$ steps of an MCMC corrector targeting $\nu_{t|z}$ to reduce discretization error. In our setting, this is particularly effective since the log-density is available, enabling Metropolis–Hastings corrections. See Subsec. H.5 for the corresponding analysis of these hyperparameters.

# 4. Related Work

We review related methods for sampling from unnormalized multimodal distributions. Since most effective approaches rely on constructing a bridge between a tractable reference distribution and the target, we organize prior work according to the mechanism used to define this bridge.

## 4.1. Annealing-based Methods

Annealing defines a sequence of intermediate densities that gradually transport samples from a reference to a target, see Eq. 2. Annealing-based methods differ in how they navigate this sequence. PT runs multiple Markov chains in parallel, periodically proposing swaps between adjacent chains to facilitate information exchange. In contrast, Annealed Importance Sampling (AIS, (Neal, 2001)) and Sequential Monte Carlo (SMC, (Del Moral et al., 2006)) propagate a population of weighted particles sequentially, with SMC additionally employing resampling to prevent mode collapse.

The performance of annealing methods is highly sensitive to the schedule and number of distributions, and can suffer from mass teleportation, where probability mass abruptly shifts between disjoint modes (Woodard et al., 2009). Optimizing the schedule and increasing its density can mitigate these issues, but performance still sharply deteriorates when adjacent distributions share little overlap (Syed et al., 2024). For challenging problems with high discrepancy between $\nu_{\text{ref}}$ and $\nu$, the number of interpolating distributions required for stable sampling can be prohibitively expensive.

While CDS also leverages PT for initialization, it targets a conditional distribution $\nu_{t_0|z}$ that concentrates near the reference for near-zero $t_0$, ensuring substantial overlap between successive distributions. This makes it easier to propagate samples from $\nu_{\text{ref}}$ to $\nu_{t_0|z}$ than directly to $\nu$.

## 4.2. Diffusion-based Methods

Recently, generative modelling has popularized another way of bridging distributions through diffusion processes. A diffusion process defines a continuous stochastic transformation that transports samples from a source distribution to a target distribution. This transformation corresponds to an SDE that gradually injects or removes noise.

**Diffusion-based Neural Samplers.** Neural Samplers amortize sampling by training neural networks to generate samples from a target distribution (Arbel et al., 2021; Midgley et al., 2023). Motivated by diffusion models, several works adapt these formulations to design new Neural Samplers (Nusken et al., 2024; He et al., 2025a; Havens et al., 2025; Akhound-Sadegh et al., 2025; Albergo & Vanden-Eijnden, 2025; Rissanen et al., 2025). These methods are trained using both target density evaluations and samples, either produced by earlier model iterations or generated via MCMC. While training is computationally expensive, inference requires only a single forward pass with no additional density evaluations. Therefore, Neural Samplers operate in a different but complementary regime from MCMC methods.

*Table 1.* Sampling tasks considered in this work. See App. E for more details.

| Task name | Target dist. | Dim. ($D$) |
|---|---|---|
| Gaussian Mixture (GM) | GM-2 | 2 |
| | GMNU-2 | 2 |
| | GM-16 | 16 |
| | GMNU-16 | 16 |
| Lennard-Jones (LJ) | LJ-13 | $3 \times 13 = 39$ |
| | LJ-55 | $3 \times 55 = 165$ |
| Alanine Dipeptide (ALDP) | ALDP | $3 \times 22 = 66$ |
| Bayesian Neural Network (BNN) | BNN | 550 |

**Non-amortized Methods.** This line of work aims to exploit diffusion-based ideas without resorting to neural network training. Diffusive Gibbs Sampling (DiGS, Chen et al. (2024)) introduces Gaussian convolutions to define a noisy auxiliary distribution that bridges isolated modes. However, it relies on a Metropolis-within-Gibbs procedure that scales poorly to high dimensions. Reverse Diffusion Monte Carlo (RDMC, Huang et al. (2024)) expresses the score of the marginal distribution in terms of expectations under the denoising posterior. To approximate this, it employs a nested MCMC procedure in which multiple samples from the denoising posterior are drawn at each iteration. Consequently, several density evaluations are required per iteration.

Both DiGS and RDMC rely on marginal probability paths with intractable scores. In contrast, the proposed CDS follows a conditional probability path whose score admits a closed-form expression, enabling efficient sampling without approximations.

# 5. Experiments

In this section, we evaluate CDS with a linear interpolant on eight target distributions across four diverse tasks. We study the impact of the initial time $t_0$ on PT communication efficiency in Subsec. 5.1 and compare CDS with state-of-the-art samplers in Subsec. 5.3. Experimental details are provided in App. G, with additional experiments in App. H[1]

**Sampling Tasks.** The tasks considered for evaluation are summarized in Tab. 1. In total, we consider eight target distributions across four tasks spanning synthetic benchmarks, physical systems, molecular dynamics, and high-dimensional Bayesian inference problems. These tasks cover multimodality, complex energy landscapes, and structured posteriors in a wide range of dimensions. Notably, for the ALDP and BNN targets, density and score evaluations are computationally expensive, making them particularly interesting for assessing sampling efficiency. Full details are provided in App. E.

---

[1]The code is available at github.com/Franblueee/conditional_diffusion_sampling.

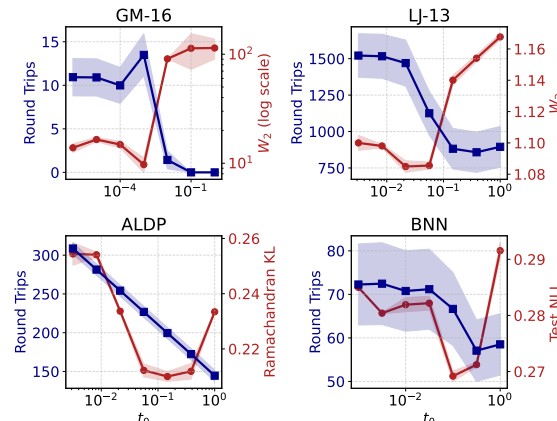

*Figure 4.* **Round Trips** (RTs, higher is better) and **sampling error** (lower is better) as a function of $t_0$. Decreasing $t_0$ from 1.0 generally increases RT counts and reduces sampling error, indicating improved mixing and sample quality.

**Sample Quality Metrics.** We evaluate sample quality using standard task-specific metrics (lower is better for all): the Wasserstein-2 ($W_2$) distance for the GM and LJ tasks; the Kullback-Leibler (KL) divergence between Ramachandran plots for ALDP; and the test negative log-likelihood (NLL) for the BNN task. Additional metrics and implementation details are provided in App. F and H.

## 5.1. Impact of Initial Time $t_0$ in Communication and Sample Quality

We analyze the communication efficiency of PT when targeting the conditional distribution $\nu_{t|z}$, and how it impacts sample quality. Recall that CDS begins by sampling $\nu_{t_0|z}$ for a chosen $t_0 > 0$. As $t \to 0$, $\nu_{t|z}$ concentrates toward a Dirac delta at $z$ (drawn from the reference distribution), increasing overlap with the reference and potentially improving communication. We measure efficiency using Round Trips (RTs), the number of traversals between reference and target distributions. Higher RTs indicate better mixing (Syed et al., 2022). We provide further details in App. H.

As shown in Fig. 4, decreasing $t_0$ from 1.0 consistently increases RTs across tasks, confirming that sampling from $\nu_{t_0|z}$ is more efficient than sampling $\nu$ directly, and leading to improved sample quality. However, as $t_0 \to 0$, both efficiency and quality deteriorate as the distribution becomes excessively concentrated, which reduces overlap between replicas. As illustrated in Fig. 3, this suggests an optimal range for $t_0$: small enough to enhance overlap with the reference, yet large enough to avoid degeneracy. A strategy for selecting $t_0$ is discussed in App. C.

## 5.2. Analysis of the Transport Mechanism

We study the impact of the SDE integration in CDS's Stage 2. After sampling from $\nu_{t_0|z}$, samples must be transported

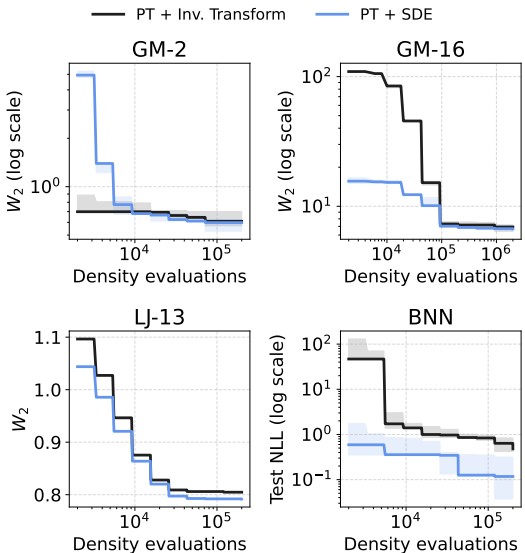

*Figure 5.* **Comparison between SDE-based transport and the inverse interpolation map.** We evaluate the effect of two strategies for transporting samples from $\nu_{t_0|z}$ back to the target distribution $\nu$. The SDE-based approach consistently achieves better performance across tasks.

to the target $\nu$. In our method, this is done by integrating the interpolation SDE in Eq. 15 from $t_0$ to 1.

Alternatively, one can apply the inverse interpolation map $F_{t_0|z}^{-1}$ to directly map samples to $\nu$. If $\nu_{t_0|z}$ were sampled perfectly, this transformation would recover perfect samples from $\nu$. In practice, however, approximation errors are amplified by the inverse map. In contrast, the SDE dynamics provide an advantage: they continuously refine the samples during transport, improving mixing and progressively correcting deviations from the target distribution.

We empirically compare CDS with this baseline, where PT is used to sample from $\nu_{t_0|z}$ followed by $F_{t_0|z}^{-1}$. Results in Fig. 5 show that SDE-based transport consistently outperforms the inverse mapping. The latter performs slightly better on GM-2 under small budgets, due to a trade-off: fewer SDE steps leave room for more exploration during the PT phase. This advantage disappears in more complex target distributions, where the corrective effect of the SDE becomes important.

### 5.3. Comparison with Other Sampling Methods

We benchmark CDS against five state-of-the-art MCMC methods, analyzing the trade-off between sample quality and total density evaluations. Full experimental details are provided in App. G.

**Baselines.** We select three recent state-of-the-art methods: Non-Reversible PT (NRPT) (Syed et al., 2022), Optimized Annealed SMC (OASMC) (Syed et al., 2024), and Diffusive Gibbs Sampling (DiGS) (Chen et al., 2024). We also include

two MCMC baselines: Hamiltonian Monte Carlo (HMC) and Metropolis–Adjusted Langevin Algorithm (MALA). We exclude RDMC from this comparison as it proved computationally intractable for the tasks considered, consistent with findings in Chen et al. (2024). In Subsec. H.3, we include a comparison against No-U-Turn Sampler (NUTS, Hoffman et al. (2014)), Stein Variational Gradient Descent (SVGD, Liu & Wang (2016)), and the Metropolis Adjusted Microcanonical Sampler (MAMS, Robnik et al. (2025)). As discussed in Subsec. 4.2, we exclude Neural Samplers as they operate in a different, complementary regime.

**Evaluation Protocol.** Sampling performance is assessed via average Pareto fronts, capturing the trade-off between sample quality and computational cost. In addition to task-specific quality metrics, we report the Mean Hypervolume Ratio (HVR, Zitzler et al. (2003)), computed on normalized Pareto fronts to enable comparability across objectives with different scales. Further details are provided in App. F.

**Results Overview.** Overall results are summarized in Table 2. CDS, with a mean HVR of 0.9863, significantly outperforms the standard MCMC baselines (HMC, MALA) and the strongest competitor, NRPT. Thus, it achieves the best trade-offs between cost and sampling accuracy.

**Per-task Analysis.** Representative trade-off curves are shown in Fig. 6, with additional results reported in App. H. On the GM task, the proposed method exhibits stable performance across dimensions. While CDS and DiGS outperform all samplers in low-dimensional settings (GM-2 and GMNU-2), DiGS performance deteriorates in the 16-dimensional cases (GM-16 and GMNU-16), likely due to the limited efficiency of its Metropolis-within-Gibbs mechanism. In contrast, CDS maintains its efficiency and achieves the best trade-offs in these medium-dimensional regimes.

CDS remains highly competitive on physical system benchmarks. On LJ, MALA and HMC achieve the best Pareto fronts (as expected, since local samplers suffice to capture the target distribution in this task). CDS performs on par with NRPT and even surpasses it in the high-density evaluation budget regime. In contrast, performance on ALDP reveals a clear separation: MALA and HMC degrade severely, while CDS rivals NRPT for the best results, followed by OASMC and DiGS. Notably, CDS remains robust despite the unfavorable interaction between the LJ potential (also present in ALDP) and the linear interpolant, which drives interparticle distances toward zero as $t \to 0$. Even under these adverse conditions, which are induced by the interpolant choice, CDS avoids catastrophic failure and performs remarkably well.

In the highest-dimensional BNN task ($D = 550$), CDS significantly outperforms all samplers, demonstrating superior capability in exploring complex, multimodal posteriors.

*Table 2.* **Aggregated Mean Hypervolume Ratio (HVR) across sampling tasks.** Higher HVR values (↑) denote superior performance in terms of both convergence and coverage of the optimal Pareto front (Zitzler et al., 2003). See App. F for details.

| Method | CDS (ours) | NRPT | OASMC | DiGS | HMC | MALA |
|---|---|---|---|---|---|---|
| HVR (↑) | **0.9976 ± 0.0015** | 0.9827 ± 0.0083 | 0.9287 ± 0.0277 | 0.5464 ± 0.1550 | 0.6263 ± 0.1261 | 0.5241 ± 0.1494 |

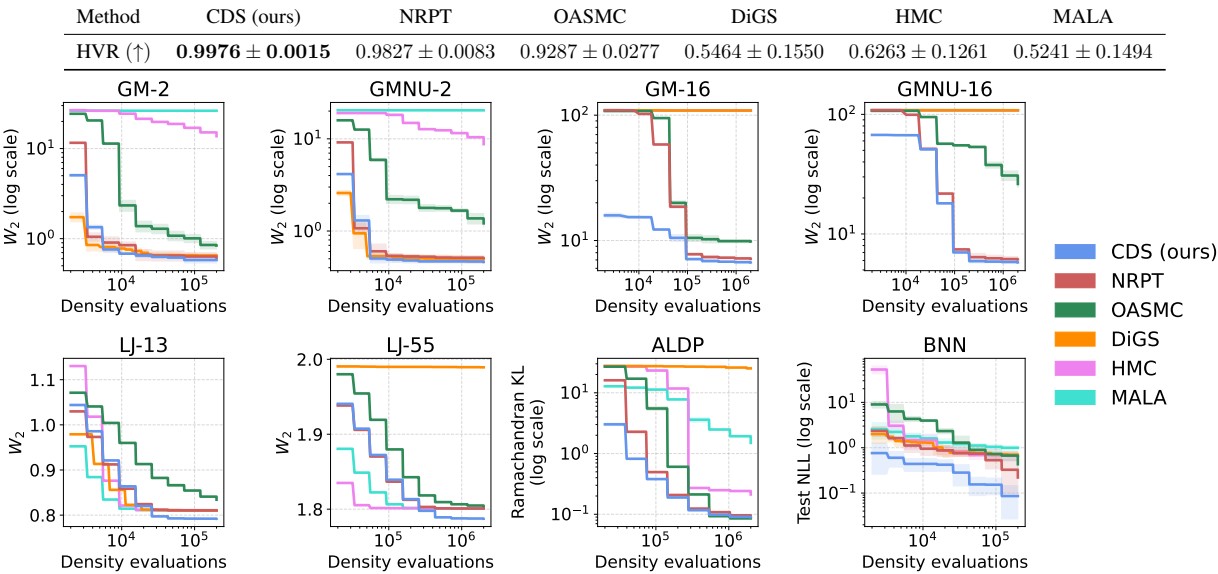

*Figure 6.* **Pareto fronts for sampling performance across eight target distributions.** The proposed CDS method achieves competitive or superior performance compared to state-of-the-art samplers, demonstrating higher efficiency by requiring fewer density evaluations for the same level of accuracy.

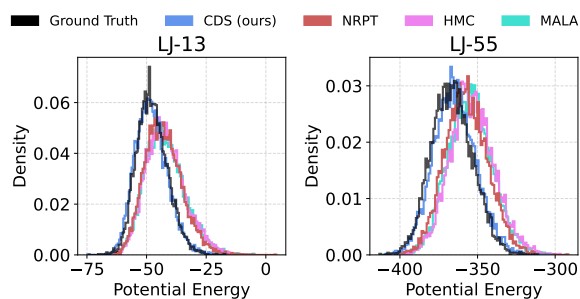

*Figure 7.* **Comparison of Lennard-Jones (LJ) potential energy histograms.** For clarity, results for the remaining methods are provided in Fig. 13. All samplers use a fixed budget of $2 \cdot 10^4$ and $2 \cdot 10^5$ density evaluations for the LJ-13 and LJ-55 targets, respectively. CDS provides the most accurate match to the ground truth histograms, closely followed by NRPT.

**Qualitative Analysis.** We analyze how quantitative performance differences translate into qualitative sample fidelity. Fig. 12 visualizes samples from the GM task using $2 \cdot 10^3$ density evaluations. Both CDS and DiGS recover all modes with correct proportions, whereas competing methods fail to discover all of them. Energy histograms for the LJ systems (Fig. 7) indicate that CDS matches the target energy distribution most accurately, closely followed by NRPT. For the ALDP task, the Ramachandran histograms (Fig. 2) show that only CDS and NRPT successfully recover the two modes with accurate density allocations, though NRPT captures the central isolated mode more effectively.

## 6. Conclusions

In this work, we introduced Conditional Diffusion Sampling (CDS), a training-free framework that combines the robust global exploration of Parallel Tempering (PT) with the efficient local transport of diffusion processes. By deriving Conditional Interpolants, we obtained exact, closed-form SDEs that transport samples from an accessible initialization distribution to the target. Both theoretical and empirical results show that the initialization cost vanishes at short diffusion times. Our extensive evaluation on high-dimensional benchmarks demonstrates that CDS achieves a superior trade-off between sample quality and computational cost.

**Limitations and Future Work.** While CDS demonstrates robust performance across varying dimensionalities, our experimental analysis highlights the following two limitations. First, our results in the LJ and ALDP tasks suggest that the choice of interpolant is critical for densities with singularities, as it may force trajectories through numerical instabilities in high-energy regions. Second, we observed that the choice of the initialization time $t_0$ involves a delicate trade-off: excessively small values cause the initialization distribution $\nu_{t_0|z}$ to become extremely peaked, hindering communication efficiency in the PT stage.

Crucially, these limitations stem primarily from the specific use of the linear interpolant rather than the CDS framework itself. This opens a promising research direction: designing better interpolants that take into account the underlying geometry of the target density to further improve performance.

## Acknowledgements

FMCM acknowledges contract FPU21/01874 (Ministerio de Universidades). PMA acknowledges grant C-EXP-153-UGR23 (Consejería de Universidad, Investigación e Innovación and European Union ERDF Andalusia Program 2021-2027). SS acknowledges support from NSERC Discover Grant RGPIN-2026-07176. DHL acknowledges support from project PID2022-139856NB-I00 (MCIN/AEI/10.13039/501100011033 and FEDER, UE), project IDEA-CM (TEC-2024/COM-89, Comunidad de Madrid), and the ELLIS Unit Madrid. JMHL acknowledges support from EPSRC funding under grant EP/Y028805/1. RMS, PMA, FMCM acknowledges support from project PID2022-140189OB-C22 (MCIN/AEI/10.13039/501100011033 and FEDER, UE).

## Impact Statement

This paper presents work whose goal is to advance the field of Machine Learning. There are many potential societal consequences of our work, none of which we feel must be specifically highlighted here.

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

# A. Extended Background: Parallel Tempering

In this section, we describe Parallel Tempering (PT), a Markov chain Monte Carlo (MCMC) method designed to sample from a complex target distribution $\nu$. We follow the exposition by Syed et al. (2022). Let $\nu_{\text{ref}}$ be a reference distribution from which independent and identically distributed (i.i.d.) sampling is tractable. In the following, we assume $\nu$ and $\nu_{\text{ref}}$ have densities $\pi$ and $\pi_{\text{ref}}$, respectively.

PT constructs a sequence of $N + 1$ distributions interpolating between the reference $\pi_{\text{ref}}$ and the target $\pi$. This is governed by an annealing schedule $\mathcal{B}_N = \{0 = \beta_0 < \beta_1 < \cdots < \beta_N = 1\}$. The distribution for the $n$-th chain, $\pi_{\beta_n}$, is typically defined as:

$$\pi_\beta(x) \propto \pi_{\text{ref}}(x)^{1-\beta} \pi(x)^\beta. \tag{20}$$

The algorithm simulates a Markov chain on the expanded state space $\mathcal{X}^{N+1}$ targeting the product distribution:

$$\Pi(\mathbf{x}) = \pi_{\beta_0}(x^0)\pi_{\beta_1}(x^1)\cdots\pi_{\beta_N}(x^N), \tag{21}$$

where $\mathbf{x} = (x^0, \ldots, x^N)$. Each iteration of the PT algorithm consists of two distinct steps: local exploration and communication.

**Local exploration.** A $\pi_{\beta_n}$-invariant Markov kernel $K_{\beta_n}$ (e.g., MALA or HMC) is applied to update the state of each chain independently. See Subsec. D.2 for an introduction to Markov kernels.

**Communication.** Swaps between the states of adjacent chains are proposed. They are accepted or rejected according to a Metropolis-Hastings criterion. The acceptance probability $\alpha_n(\mathbf{x})$ is given by:

$$\alpha_n(\mathbf{x}) = \max\left\{1, \frac{\pi_{\beta_n}(x^{n+1})\pi_{\beta_{n+1}}(x^n)}{\pi_{\beta_n}(x^n)\pi_{\beta_{n+1}}(x^{n+1})}\right\}. \tag{22}$$

Efficient communication implies that these swaps are frequently accepted, allowing samples from the tractable $\nu_{\text{ref}}$ to traverse the path to $\nu$. Non-reversible PT variants alternate between swaps of odd chains and swaps of even chains. They have proven to outperform their reversible counterparts (Syed et al., 2022). The pseudocode for Non-Reversible PT (NRPT) is shown in Alg. 2.

---

**Algorithm 2** Non-Reversible Parallel Tempering (NRPT)

---

**Require:** Number of iterations $K$, annealing schedule $\mathcal{B}_N$.
1: Initialize $\mathbf{x}_0 = (x_0^0, x_0^1, \ldots, x_0^N)$ with $x_0^n \sim \pi_{\text{ref}}$.
2: **for** $k = 1$ to $K$ **do**
3:     // Local Exploration
4:     **for** $n = 0$ to $N$ **do**
5:         $x_k^n \leftarrow$ Sample from $K_{\beta_n}(x_{k-1}^n, \cdot)$.
6:     **end for**
7:     // Communication (Swaps)
8:     **for** $n \equiv k \mod 2$ **do**
9:         $\alpha_n \leftarrow \alpha_n(\mathbf{x}_k)$ // See Eq. 22
10:         Draw $u \sim \mathcal{U}(0, 1)$
11:         **if** $u < \alpha_n$ **then**
12:             $x_k^{n+1}, x_k^n \leftarrow x_k^n, x_k^{n+1}$
13:         **end if**
14:     **end for**
15: **end for**
16: **return** $(\mathbf{x}_1, \ldots, \mathbf{x}_K)$

---

**Round Trips.** A round trip is defined as the event where a specific particle, initialized at the reference distribution (index 0), traverses up to the target distribution (index $N$) and returns to the reference. The round trip rate $\tau_N$ is the frequency of round trips over $K$ iterations. A higher round trip rate indicates that the chains are exchanging information efficiently and that the target distribution is being effectively explored by samples originating from the reference.

**Global Communication Barrier (GCB).** The performance of PT can be analyzed in the asymptotic limit where the number of chains $N \to \infty$. In this regime, the efficiency is characterized by the Global Communication Barrier (GCB), denoted $\Lambda(\nu_{\text{ref}}, \nu)$. It is defined as the accumulated local communication barriers across the path:

$$\Lambda(\nu_{\text{ref}}, \nu) = \frac{1}{2} \int_0^1 \lambda(\beta) \, \mathrm{d}\beta, \quad \lambda(\beta) = \int |l(x) - l(y)| \, \pi_\beta(x)\pi_\beta(y) \, \mathrm{d}x \, \mathrm{d}y, \tag{23}$$

where $l(\cdot)$ is the log-likelihood ratio $l(\cdot) = \log \pi(\cdot) - \log \pi_{\text{ref}}(\cdot)$. The asymptotic round trip rate converges to a value determined by the GCB:

$$\lim_{N \to \infty} \tau_N = \frac{1}{2 + 2\Lambda(\nu_{\text{ref}}, \nu)}. \tag{24}$$

Consequently, a lower GCB indicates higher communication efficiency and a faster restart rate. From (Surjanovic et al., 2022), the GCB is upper bounded by the square root of the symmetric KL divergence:

$$\Lambda(\nu_{\text{ref}}, \nu) \leq \sqrt{\frac{1}{2}\text{SKL}(\nu_{\text{ref}}, \nu)}, \tag{25}$$

where $\text{SKL}(\nu_{\text{ref}}, \nu) = \text{KL}(\nu_{\text{ref}}, \nu) + \text{KL}(\nu, \nu_{\text{ref}})$.

## B. Conditional Flows

**Flows.** When the diffusion coefficient in Eq. 3 vanishes, $\sigma_t = 0$, the SDE in Eq. 3 reduces to the ordinary differential equation (ODE):

$$\mathrm{d}x_t = u_t(x_t) \, \mathrm{d}t, \tag{26}$$

where $(t, x) \mapsto a_t(x) \in \mathbb{R}^D$ is the velocity vector field. In this case, the FPK equation Eq. 4 degenerates into the continuity equation

$$\frac{\partial}{\partial t}p_t = -\text{div}(p_t u_t), \tag{27}$$

which characterizes the evolution of densities transported by the deterministic flow induced by $u_t$. As in diffusions, one can choose the velocity field $u_t$ such that $p_0 = \pi_{\text{ref}}$, and $p_1 = \pi$ (Lipman et al., 2023). Thus, the ODE Eq. 26 defines a continuous probabilistic bridge between $\pi_{\text{ref}}$ and $\pi$.

**The Conditional ODE.** For a reference variable $z \sim \nu_{\text{ref}}$, the interpolant defined in Eq. 7 describes a time-dependent probability path. This path can be characterized by an ODE associated with the following velocity field:

$$\mathrm{d}x_t = u_{t|z}(x_t) \, \mathrm{d}t, \tag{28}$$

$$u_{t|z}(x) = \frac{\partial F_{t|z}}{\partial t}\left(F_{t|z}^{-1}(x)\right). \tag{29}$$

This vector field is well-defined for $t \in (0, 1]$. By construction, if $\{x_t\}$ solves this ODE, its marginal density at time $t$ is exactly $\pi_{t|z}$, as the density path satisfies the associated continuity equation, see Proposition D.4.

> *Example* B.1. For the linear interpolant introduced in Eq. 6, the vector field takes the simple form:
>
> $$u_{t|z}(x) = \frac{x - z}{t}. \tag{30}$$

The boundary conditions defining the interpolant make it impossible for $F_{t|z}$ to be invertible at $t = 0$. As a consequence, the vector field $u_{t|z}$ generally exhibits a singularity at $t = 0$, causing solutions to diverge as $t \to 0$. This mirrors the boundary singularities observed in diffusion models and flow matching, where vanishing noise variance leads to unbounded vector fields or scores (Song et al., 2021; Lipman et al., 2023). Consequently, numerical integration cannot proceed from $t = 0$. Instead, one must initialize the process at a initial time $t_0 > 0$ with a sample $x_{t_0} \sim \nu_{t_0|z}$. Crucially, for the linear interpolant, the trajectory for $t > t_0$ is entirely deterministic given $x_{t_0}$ and $z$, as it is given by the integration of the velocity field in Eq. 26. This reveals a limitation of the ODE formulation: the target distribution $\nu$ influences the process *only* through the initial sample $x_{t_0}$. Once initialized, the dynamics do not utilize the target information to refine the trajectory.

## C. Optimizing the Initial Time $t_0$

A critical design choice within the proposed CDS framework is the selection of the initial time instant, $t_0$, used to sample from $\nu_{t_0|z}$ and initialize the integration. In this section, we propose an optimization-based approach to determine this value.

We propose to minimize the Symmetric Kullback-Leibler (SKL) divergence between the reference distribution $\nu_{\text{ref}}$ and the target $\nu_{t|z}$ with respect to $t$. The SKL is defined as:

$$\text{SKL}\left(\nu_{\text{ref}}, \nu_{t|z}\right) = \text{KL}\left(\nu_{\text{ref}}, \nu_{t|z}\right) + \text{KL}\left(\nu_{t|z}, \nu_{\text{ref}}\right), \tag{31}$$

where $\text{KL}(\cdot, \cdot)$ denotes the standard Kullback-Leibler divergence. Consequently, we define the optimal start time $t_0$ as:

$$t_0 = \arg\min_t \text{SKL}\left(\nu_{\text{ref}}, \nu_{t|z}\right). \tag{32}$$

This criterion is motivated by the relationship between the SKL and the Global Communication Barrier (GCB), denoted as $\Lambda(\cdot, \cdot)$. As shown in (Surjanovic et al., 2022), the SKL provides an upper bound on the GCB:

$$\Lambda\left(\nu_{\text{ref}}, \nu_{t|z}\right) \leq \sqrt{\frac{1}{2}\text{SKL}\left(\nu_{\text{ref}}, \nu_{t|z}\right)}. \tag{33}$$

In annealing-based methods such as Parallel Tempering (PT) and Optimized Sequential Monte Carlo (OSMC) (Syed et al., 2021; 2024), the GCB serves as a global measure of transport difficulty along the annealing path. Lower values imply easier communication and higher sampling efficiency. Since direct minimization of the GCB is often intractable, the SKL serves as an effective and differentiable proxy (Surjanovic et al., 2022).

In CDS, when PT or OSMC is used in the first stage, minimizing this objective amounts to maximizing the communication efficiency of PT or OSMC. Next, we obtain closed-form expressions for the derivative of SKL with respect to $t$ and propose an online optimization strategy.

### C.1. Online optimization

Since the SKL divergence is differentiable with respect to $t$, we can employ stochastic optimization to find the optimal value $t_0$. The gradient with respect to $t$ is given below.

**Lemma C.1.** *The derivative of the SKL with respect to $t$ is given by:*

$$\frac{\partial}{\partial t}\text{SKL}(\nu_{ref}, \nu_{t|z}) = \mathbb{E}_{x \sim \nu_{t|z}}\left[\left(\frac{\partial}{\partial t}\log\pi_{t|z}(x)\right)\log\frac{\pi_{t|z}(x)}{\pi_{ref}(x)}\right] - \mathbb{E}_{x \sim \nu_{ref}}\left[\frac{\partial}{\partial t}\log\pi_{t|z}(x)\right]. \tag{34}$$

*Proof.* Let $J(t) = \text{SKL}(\nu_{\text{ref}}, \nu_{t|z})$. We decompose this into the forward and reverse KL divergences:

$$J(t) = \underbrace{\int \pi_{t|z}(x)\log\frac{\pi_{t|z}(x)}{\pi_{\text{ref}}(x)}\,dx}_{A_t} + \underbrace{\int \pi_{\text{ref}}(x)\log\frac{\pi_{\text{ref}}(x)}{\pi_{t|z}(x)}\,dx}_{B_t}. \tag{35}$$

For $A_t$, we differentiate under the integral sign and use the identity $\frac{\partial}{\partial t}\pi_{t|z} = \pi_{t|z}\frac{\partial}{\partial t}\log\pi_{t|z}$:

$$\frac{\partial}{\partial t}A_t = \int \pi_{t|z}(x)\left(\frac{\partial}{\partial t}\log\pi_{t|z}(x)\right)\log\frac{\pi_{t|z}(x)}{\pi_{\text{ref}}(x)}\,dx + \underbrace{\int \pi_{t|z}(x)\frac{\partial}{\partial t}\log\pi_{t|z}(x)\,dx}_{=\frac{\partial}{\partial t}\int \pi_{t|z}(x)dx = 0}. \tag{36}$$

Thus, $\frac{\partial}{\partial t}A_t = \mathbb{E}_{x \sim \nu_{t|z}}\left[\left(\frac{\partial}{\partial t}\log\pi_{t|z}(x)\right)\log\frac{\pi_{t|z}(x)}{\pi_{\text{ref}}(x)}\right]$.

For $B_t$, since $\pi_{\text{ref}}$ is independent of $t$:

$$\frac{\partial}{\partial t}B_t = \int \pi_{\text{ref}}(x)\frac{\partial}{\partial t}\left(\log\pi_{\text{ref}}(x) - \log\pi_{t|z}(x)\right)dx = -\int \pi_{\text{ref}}(x)\frac{\partial}{\partial t}\log\pi_{t|z}(x)\,dx. \tag{37}$$

Thus, $\frac{\partial}{\partial t}B_t = -\mathbb{E}_{x \sim \nu_{\text{ref}}}\left[\frac{\partial}{\partial t}\log\pi_{t|z}(x)\right]$. Summing the derivatives of $A_t$ and $B_t$ yields the result. $\qquad\square$

*Example* C.2. In the case of the linear interpolant, the derivative of SKL with respect to $t$ is given by

$$\frac{\partial}{\partial t} \mathrm{SKL}(\nu_{\mathrm{ref}}, \nu_{t|z}) = \mathbb{E}_{x \sim \nu_{t|z}} \left[ s_{t|z}(x) \log \frac{\pi_{t|z}(x)}{\pi_{\mathrm{ref}}(x)} \right] - \mathbb{E}_{x \sim \nu_{\mathrm{ref}}} \left[ s_{t|z}(x) \right]. \tag{38}$$

where $s_{t|z}(x) = -\frac{1}{t^2}(x - z)^\top \nabla \log \pi_{t|z}(x) - \frac{d}{t}$.

In practice, this gradient can be used within an online stochastic gradient descent scheme. Starting from an initial guess $t_0$, we run the first-stage sampler (e.g., PT or OASMC) and use the generated particles from $\nu_{t_0|z}$ to estimate the expectations and update $t$ iteratively. This optimization can be performed during the burn-in phase of the initial sampler and continued until the first stage of CDS is completed. Importantly, this procedure incurs no additional target density evaluations, as all required gradients and density values can be cached for each particle.

## D. Theoretical Results

### D.1. Conditional Interpolants

**Setting.** Let $F \colon [0,1] \times \mathcal{X} \times \mathcal{X} \to \mathcal{X}$ be a differentiable map denoted as $F_t(z, x) = F(t, z, x)$, satisfying the boundary conditions $F_0(z, x) = z$ and $F_1(z, x) = x$. For a fixed $z \in \mathcal{X}$, we define the map $F_{t|z} \colon \mathcal{X} \to \mathcal{X}$ as

$$F_{t|z}(x) = F_t(z, x). \tag{39}$$

We assume that $F_{t|z}$ is a diffeomorphism (i.e., invertible with a differentiable inverse).

Let $\nu$ be a probability measure defined on $\mathcal{X}$. For a fixed $z \in \mathcal{X}$, define the interpolated process:

$$x_t = F_{t|z}(x), \quad x \sim \nu. \tag{40}$$

Let $\nu_{t|z}$ denote the distribution of the random variable $x_t$ conditioned on $z$. Formally, this is the pushforward measure:

$$\nu_{t|z} = F_{t|z} \# \nu. \tag{41}$$

**Lemma D.1** (Convergence to Dirac mass). *Assume that $\mathcal{X}$ is a separable normed vector space. Assume the following regularity conditions:*

1. *$\nu$ has a finite first-order moment.*

2. *For $\nu$-almost all $x$, $\lim_{t \to 0} \| F_{t|z}(x) - z \| = 0$.*

3. *There exists an integrable function $g : \mathcal{X} \to [0, +\infty)$ such that for all sufficiently small $t$, $\| F_{t|z}(x) - z \| \le g(x)$.*

*Then, the conditional distribution $\nu_{t|z}$ converges to the Dirac mass $\delta_z$ in the Wasserstein-1 distance:*

$$\lim_{t \to 0} W_1(\delta_z, \nu_{t|z}) = 0, \tag{42}$$

*Proof.* In any normed space, the Wasserstein-1 distance between a Dirac mass $\delta_z$ and an arbitrary probability measure $\mu$ is given exactly by the expected distance to $z$:

$$W_1(\delta_z, \mu) = \int_{\mathcal{X}} \| z - y \| \, \mu(\mathrm{d}y). \tag{43}$$

Substituting $\mu = \nu_{t|z}$ and applying the change of variables formula:

$$W_1(\delta_z, \nu_{t|z}) = \int_{\mathcal{X}} \| z - F_{t|z}(x) \| \, \nu(\mathrm{d}x). \tag{44}$$

We analyze the limit of this integral as $t \to 0$. By the first regularity condition, the integrand converges to 0 for almost all $x$. By the second regularity condition, the integrand is bounded by the integrable function $g(x)$. Therefore, by the Dominated Convergence Theorem:

$$\lim_{t \to 0} \int_{\mathcal{X}} \| z - F_{t|z}(x) \| \, \mu(\mathrm{d}x) = 0. \tag{45}$$

$\square$

**Lemma D.2** (Conditional Density). *Let $\pi$ be the density of $\nu$. Then, the density of $\nu_{t|z}$ is given by:*

$$\pi_{t|z}(x) = \left|\det \mathrm{J}F_{t|z}\left(F_{t|z}^{-1}(x)\right)\right|^{-1} \pi\left(F_{t|z}^{-1}(x)\right), \tag{46}$$

*where $\mathrm{J}F_{t|z}$ denotes the Jacobian matrix of the map $F_{t|z}$.*

*Proof.* Let $A \subseteq \mathcal{X}$ be an arbitrary measurable set. Let $\pi_{t|z}$ be the density of $\nu_{t|z}$. By the definition of the pushforward measure, we relate the integrals over the densities as follows:

$$\int_A \pi_{t|z}(x)\,\mathrm{d}x = \nu_{t|z}(A) = \nu\left(F_{t|z}^{-1}(A)\right) = \int_{F_{t|z}^{-1}(A)} \pi(y)\,\mathrm{d}y. \tag{47}$$

We apply the change of variables formula to the integral on the right-hand side. Let $x = F_{t|z}(y)$. By the Inverse Function Theorem, it holds:

$$\left|\det \mathrm{J}(F_{t|z}^{-1})(x)\right| = \left|\det\left(\mathrm{J}F_{t|z}(F_{t|z}^{-1}(x))\right)^{-1}\right| = \left|\det \mathrm{J}F_{t|z}(F_{t|z}^{-1}(x))\right|^{-1}. \tag{48}$$

Substituting this back into the integral and applying the change of variables formula, we obtain:

$$\int_A \pi_{t|z}(x)\,\mathrm{d}x = \int_{F_{t|z}^{-1}(A)} \pi(y)\,\mathrm{d}y = \int_A \pi\left(F_{t|z}^{-1}(x)\right)\left|\det \mathrm{J}F_{t|z}\left(F_{t|z}^{-1}(x)\right)\right|^{-1}\mathrm{d}x. \tag{49}$$

Since this equality holds for any measurable set $A$, the integrands must be equal almost everywhere. Therefore:

$$\pi_{t|z}(x) = \pi\left(F_{t|z}^{-1}(x)\right)\left|\det \mathrm{J}F_{t|z}\left(F_{t|z}^{-1}(x)\right)\right|^{-1}. \tag{50}$$

$\square$

**Lemma D.3** (Conditional Score). *Let $\pi$ be the density of $\nu$, and assume it to be differentiable and strictly positive. Then, $\pi_{t|z}$ is differentiable and its score is given by:*

$$\nabla_x \log \pi_{t|z}(x) = \left[\mathrm{J}F_{t|z}\left(F_{t|z}^{-1}(x)\right)\right]^{-\top} \nabla \log \pi\left(F_{t|z}^{-1}(x)\right) - \nabla_x \log \left|\det \mathrm{J}F_{t|z}(x)\right|. \tag{51}$$

*Proof.* From the result of the previous Lemma, the density $\pi_{t|z}$ is given by:

$$\pi_{t|z}(x) = \pi\left(F_{t|z}^{-1}(x)\right)\left|\det \mathrm{J}F_{t|z}\left(F_{t|z}^{-1}(x)\right)\right|^{-1}. \tag{52}$$

Taking the logarithm of both sides, we obtain:

$$\log \pi_{t|z}(x) = \log \pi\left(F_{t|z}^{-1}(x)\right) - \log \left|\det \mathrm{J}F_{t|z}\left(F_{t|z}^{-1}(x)\right)\right|. \tag{53}$$

We now compute the gradient with respect to $x$, denoted by $\nabla_x$. Applying the gradient operator to the equation gives:

$$\nabla_x \log \pi_{t|z}(x) = \nabla_x\left[\log \pi\left(F_{t|z}^{-1}(x)\right)\right] - \nabla_x\left[\log \left|\det \mathrm{J}F_{t|z}\left(F_{t|z}^{-1}(x)\right)\right|\right]. \tag{54}$$

All we need to do is to develop the right-hand side. Let $y = F_{t|z}^{-1}(x)$. By the chain rule:

$$\nabla_x\left[\log \pi\left(F_{t|z}^{-1}(x)\right)\right] = \left(\mathrm{J}F_{t|z}^{-1}(x)\right)^{\top} \nabla_y \log \pi(y)\Big|_{y=F_{t|z}^{-1}(x)}. \tag{55}$$

The Inverse Function Theorem yields:

$$\mathrm{J}F_{t|z}^{-1}(x) = \left[\mathrm{J}F_{t|z}\left(F_{t|z}^{-1}(x)\right)\right]^{-1}. \tag{56}$$

Substituting this back into the gradient expression and using the property $(A^{-1})^{\top} = A^{-\top}$:

$$\nabla_x\left[\log \pi\left(F_{t|z}^{-1}(x)\right)\right] = \left[\mathrm{J}F_{t|z}\left(F_{t|z}^{-1}(x)\right)\right]^{-\top} \nabla \log \pi\left(F_{t|z}^{-1}(x)\right). \tag{57}$$

$\square$

**Proposition D.4** (Continuity and FPK Equations). *Assume that $F$ is $C^1$ in $t$ and $C^2$ in $x$. Let $u_{t|z} \colon \mathcal{X} \to \mathcal{X}$ be defined by:*

$$u_{t|z}(x) = \left( \frac{\partial F_{t|z}}{\partial t} \right) \left( F_{t|z}^{-1}(x) \right). \tag{58}$$

1. *The density $\pi_{t|z}$ satisfies the following continuity equation:*

$$\frac{\partial}{\partial t} \pi_{t|z} = -\operatorname{div}\left( \pi_{t|z} u_{t|z} \right), \tag{59}$$

2. *Furthermore, $\pi_{t|z}$ satisfies the following FPK equation:*

$$\frac{\partial}{\partial t} \pi_{t|z} = -\operatorname{div}\left( \pi_{t|z} a_{t|z} \right) + \frac{\sigma_t^2}{2} \Delta \pi_{t|z}, \tag{60}$$

*where $a_{t|z} = u_{t|z} + \frac{\sigma_t^2}{2} \nabla \log \pi_{t|z}$.*

*Proof.*

1. First, we observe that, as a consequence of change of variables formula, the following identity holds for any $y \in \mathcal{X}$:

$$\pi_{t|z}\left( F_{t|z}(y) \right) J_t(y) = \pi(y), \tag{61}$$

where $J_t(y) = \det \mathrm{J} F_{t|z}(y)$. We differentiate with respect to $t$:

$$\left[ \frac{\partial}{\partial t} \left( \pi_{t|z} \circ F_{t|z} \right)(y) \right] J_t(y) + \pi_{t|z}\left( F_{t|z}(y) \right) \left[ \frac{\partial}{\partial t} J_t(y) \right] = 0. \tag{62}$$

We analyze the first term. Applying the chain rule:

$$\frac{\partial}{\partial t} \left( \pi_{t|z} \circ F_{t|z} \right)(y) = \frac{\partial \pi_{t|z}}{\partial t}\left( F_{t|z}(y) \right) + \nabla \pi_{t|z}\left( F_{t|z}(y) \right) \cdot u_{t|z}\left( F_{t|z}(y) \right), \tag{63}$$

where we have used $\frac{\partial F_{t|z}}{\partial t}(y) = u_{t|z}(F_{t|z}(y))$. For the second term, we use Jacobi's formula for the derivative of a determinant ([Petersen et al., 2008](#)):

$$\frac{\partial}{\partial t} \det(A_t) = \det(A_t) \operatorname{tr}\left( A_t^{-1} \frac{\partial}{\partial t} A_t \right) \tag{64}$$

Taking $A_t = \mathrm{J} F_{t|z}$, we obtain:

$$\frac{\partial}{\partial t} J_t(y) = J_t(y) \operatorname{div}\left( u_{t|z}\left( F_{t|z}(y) \right) \right). \tag{65}$$

Substituting these into [Eq. 62](#), and observing that $J_t(y) \neq 0$, we obtain:

$$\left( \frac{\partial \pi_{t|z}}{\partial t}\left( F_{t|z}(y) \right) + \nabla \pi_{t|z}\left( F_{t|z}(y) \right) \cdot u_{t|z}\left( F_{t|z}(y) \right) \right) + \pi_{t|z}\left( F_{t|z}(y) \right) \operatorname{div}\left( u_{t|z}\left( F_{t|z}(y) \right) \right) = 0. \tag{66}$$

Now, let $x \in \mathcal{X}$, and $y = F_{t|z}^{-1}(x)$. The above equation becomes:

$$\left( \frac{\partial \pi_{t|z}}{\partial t}(x) + \nabla \pi_{t|z}(x) \cdot u_{t|z}(x) \right) + \pi_{t|z}(x) \operatorname{div}\left( u_{t|z}(x) \right) = 0. \tag{67}$$

Using the identity $\operatorname{div}(\pi u) = \nabla \pi \cdot u + \pi \operatorname{div}(u)$, we conclude:

$$\frac{\partial \pi_{t|z}}{\partial t}(x) + \operatorname{div}\left( \pi_{t|z} u_{t|z} \right)(x) = 0. \tag{68}$$

2. We start from the continuity equation derived above and add and subtract the term $\frac{\sigma_t^2}{2}\Delta\pi_{t|z}$ to the right-hand side:

$$\frac{\partial\pi_{t|z}}{\partial t} = -\operatorname{div}\left(\pi_{t|z}u_{t|z}\right) - \frac{\sigma_t^2}{2}\Delta\pi_{t|z} + \frac{\sigma_t^2}{2}\Delta\pi_{t|z}. \tag{69}$$

Using $\Delta\pi = \operatorname{div}(\nabla\pi)$, we obtain:

$$\frac{\partial\pi_{t|z}}{\partial t} = -\operatorname{div}\left(\pi_{t|z}u_{t|z} + \frac{\sigma_t^2}{2}\nabla\pi_{t|z}\right) + \frac{\sigma_t^2}{2}\Delta\pi_{t|z}. \tag{70}$$

Next, we use $\nabla\pi_{t|z} = \pi_{t|z}\nabla\log\pi_{t|z}$ to rewrite the term inside the divergence:

$$\pi_{t|z}u_{t|z} + \frac{\sigma_t^2}{2}\nabla\pi_{t|z} = \pi_{t|z}\left(u_{t|z} + \frac{\sigma_t^2}{2}\nabla\log\pi_{t|z}\right), \tag{71}$$

which yields the desired equality by substituting the definition of the drift $a_{t|z} = u_{t|z} + \frac{\sigma_t^2}{2}\nabla\log\pi_{t|z}$.

$\square$

## D.2. Markov Kernels and Bijections

**Markov Kernels.** We follow the exposition by Ollivier (2009). Let $(\mathcal{X}, d)$ be a complete separable metric space equipped with its Borel $\sigma$-algebra $\mathcal{B}(\mathcal{X})$. We denote by $\mathcal{P}(\mathcal{X})$ the set of probability measures on $\mathcal{X}$.

A Markov transition kernel is a map $K : \mathcal{X} \times \mathcal{B}(\mathcal{X}) \to [0, 1]$ satisfying:

1. For every $x \in \mathcal{X}$, the map $A \mapsto K(x, A)$ is a probability measure on $\mathcal{X}$.

2. For every $A \in \mathcal{B}(\mathcal{X})$, the map $x \mapsto K(x, A)$ is measurable.

We define the action of the kernel $K$ on a measure $\mu \in \mathcal{P}(\mathcal{X})$ (from the left) as the measure $K(\mu)$ given by:

$$[K(\mu)](A) = \int_{\mathcal{X}} K(x, A)\,\mathrm{d}\mu(x), \quad \text{for all } A \in \mathcal{B}(\mathcal{X}). \tag{72}$$

We define the $n$-step transition kernel $K^n$ recursively by $K^1 = K$ and $K^n(x, A) = \int_{\mathcal{X}} K(y, A)K^{n-1}(x, \mathrm{d}y)$. Consistent with the operator notation, $K^n(\mu)$ denotes the measure obtained by applying the kernel $n$ times.

A probability measure $\nu$ is said to be invariant with respect to $K$ if $K(\nu) = \nu$.

**Wasserstein Distance.** For any two probability measures $\mu, \nu \in \mathcal{P}(\mathcal{X})$, the $L^1$-Wasserstein distance $W_1(\mu, \nu)$ is defined by

$$W_1(\mu, \nu) = \inf_{\pi\in\Pi(\mu,\nu)} \int_{\mathcal{X}\times\mathcal{X}} d(x, y)\,\mathrm{d}\pi(x, y), \tag{73}$$

where $\Pi(\mu, \nu)$ is the set of all couplings of $\mu$ and $\nu$ (i.e., measures on $\mathcal{X} \times \mathcal{X}$ with marginals $\mu$ and $\nu$).

**Proposition D.5** (Markov Kernels and Bi-Lipschitz Bijections). *Let $K$ be a Markov transition kernel invariant with respect to a probability measure $\nu$. Let $F : \mathcal{X} \to \mathcal{X}$ be a bijection such that both $F$ and $F^{-1}$ are Lipschitz continuous. We denote the Lipschitz constants of $F$ and $F^{-1}$ by $L_F$ and $L_{F^{-1}}$, respectively*

*Define the pushforward kernel $K_F$ by:*

$$K_F(\hat{x}, \hat{A}) = K\left(F^{-1}(\hat{x}), F^{-1}(\hat{A})\right). \tag{74}$$

*Let $\nu_F = F\#\nu$ be the pushforward of the invariant measure. Let $z \in \mathcal{X}$ be an arbitrary starting point, and let $x_0 = F^{-1}(z)$ be its preimage in the original space.*

*The following properties hold:*

*1. (Invariant measure) The pushforward kernel $K_F$ is invariant with respect to $\nu_F$.*

*2. (Algebraic Iteration) For any measure $\mu$ and $n \geq 1$:*

$$K_F^n (\mu) = F\# \left( K^n \left( F^{-1} \# \mu \right) \right). \tag{75}$$

*3. (Wasserstein Lipschitz Bound) For any two measures $\mu_1, \mu_2$:*

$$W_1 \left( F\#\mu_1, F\#\mu_2 \right) \leq L_F \, W_1 \left( \mu_1, \mu_2 \right). \tag{76}$$

*4. (Kernel Wasserstein Relation)*

$$W_1 \left( K_F^n \left( \delta_z \right), \nu_F \right) \leq L_F \, W_1 \left( K^n \left( \delta_{x_0} \right), \nu \right). \tag{77}$$

*5. (Convergence Bounds) If there exists $\rho \in (0, 1)$ such that $W_1(K^n(\delta_x), \nu) \leq \rho^n W_1(\delta_x, \nu)$ for all $x$, then:*

$$W_1 \left( K_F^n \left( \delta_z \right), \nu_F \right) \leq L_F \, \rho^n \, W_1 \left( \delta_{x_0}, \nu \right), \tag{78}$$

$$W_1 \left( K_F^n \left( \delta_z \right), \nu_F \right) \leq L_F L_{F^{-1}} \rho^n \, W_1 \left( \delta_z, \nu_F \right). \tag{79}$$

*Proof.* We prove each item separately:

1. (Invariant measure) We first verify that $K_F$ is invariant with respect to $\nu_F$.

$$K_F \left( \nu_F \right) \left( \hat{A} \right) = \int K_F \left( \hat{x}, \hat{A} \right) \nu_F(\mathrm{d}\hat{x}) = \int K \left( F^{-1} \left( \hat{x} \right), F^{-1} \left( \hat{A} \right) \right) \left( F\#\nu \right) (\mathrm{d}\hat{x}) \overset{\text{(i)}}{=} \tag{80}$$

$$\overset{\text{(i)}}{=} \int K \left( x, F^{-1} \left( \hat{A} \right) \right) \nu(\mathrm{d}x) \overset{\text{(ii)}}{=} \nu \left( F^{-1} \left( \hat{A} \right) \right) = \left( F\#\nu \right) \left( \hat{A} \right) = \nu_F \left( \hat{A} \right), \tag{81}$$

where (i) is due to the change of variables theorem ($\hat{x} = F(x)$), and (ii) is due to $K$ being invariant with respect to $\nu$.

2. (Algebraic Iteration) We prove this by induction. For $n = 1$, observe that:

$$\left( K_F \left( \mu \right) \right) \left( \hat{A} \right) = \int K_F \left( \hat{x}, \hat{A} \right) \mu \left( \mathrm{d}\hat{x} \right) = \int K \left( F^{-1} \left( \hat{x} \right), F^{-1} \left( \hat{A} \right) \right) \mu \left( \mathrm{d}\hat{x} \right) \overset{\text{(i)}}{=}$$

$$\overset{\text{(i)}}{=} \int K \left( x, F^{-1} \left( \hat{A} \right) \right) \left( F^{-1}\#\mu \right) (\mathrm{d}x) = K \left( F^{-1}\#\mu \right) \left( F^{-1} \left( \hat{A} \right) \right) =$$

$$= \left( F\# \left( K \left( F^{-1}\#\mu \right) \right) \right) \left( \hat{A} \right),$$

where (i) follows from the change of variables. Suppose it is true for $n \geq 1$. Then:

$$K_F^{(n+1)} \left( \mu \right) = K_F \left( K_F^{(n)} \left( \mu \right) \right) = K_F \left( F\#K^{(n)} \left( F^{-1}\#\mu \right) \right) =$$

$$= F\# \left( K \left( F^{-1}\#F\#K^{(n)} \left( F^{-1}\#\mu \right) \right) \right) =$$

$$= F\# \left( K^{(n+1)} \left( F^{-1}\#\mu \right) \right).$$

3. (Wasserstein Lipschitz Bound) Let $\gamma \in \Gamma \left( \mu_1, \mu_2 \right)$ be an optimal coupling for $W_1(\mu_1, \mu_2)$. Define the pushforward coupling $\hat{\gamma} = (F, F) \#\gamma$. We first confirm that $\hat{\gamma} \in \Gamma \left( F\#\mu_1, F\#\mu_2 \right)$:

$$\hat{\gamma} \left( \mathbb{R}^d \times \hat{A} \right) = \gamma \left( \mathbb{R}^d \times F^{-1} \left( \hat{A} \right) \right) = \mu_2 \left( F^{-1} \left( \hat{A} \right) \right) = \left( F\#\mu_2 \right) \left( \hat{A} \right),$$

$$\hat{\gamma} \left( \hat{A} \times \mathbb{R}^d \right) = \gamma \left( F^{-1} \left( \hat{A} \right) \times \mathbb{R}^d \right) = \mu_1 \left( F^{-1} \left( \hat{A} \right) \right) = \left( F\#\mu_1 \right) \left( \hat{A} \right).$$

Next, we bound the transport cost using the Lipschitz property of $F$:

$$W_1(F\#\mu_1, F\#\mu_2) \leq \int \| \hat{x} - \hat{y} \| \, \hat{\gamma} \left( \mathrm{d}\hat{x}, \mathrm{d}\hat{y} \right) = \int \| F \left( x \right) - F \left( y \right) \| \, \gamma \left( \mathrm{d}x, \mathrm{d}y \right)$$

$$\leq \int L_F \, \| x - y \| \, \gamma \left( \mathrm{d}x, \mathrm{d}y \right) = L_F W_1 \left( \mu_1, \mu_2 \right).$$

4. (Kernel Wasserstein Relation) We apply the previous two results. Using Property 1 with $\mu = \delta_z$:

$$K_F^n(\delta_z) = F\# \left( K^n(F^{-1}\#\delta_z) \right) = F\# \left( K^n(\delta_{x_0}) \right), \tag{82}$$

where $x_0 = F^{-1}(z)$. Also recall $\nu_F = F\#\nu$. Now apply Property 2 with $\mu_1 = K^n(\delta_{x_0})$ and $\mu_2 = \nu$:

$$\begin{aligned} W_1 \left( K_F^n \left( \delta_z \right), \nu_F \right) &= W_1 \left( F\#K^n(\delta_{x_0}), F\#\nu \right) \\ &\leq L_F W_1 \left( K^n(\delta_{x_0}), \nu \right). \end{aligned}$$

5. (Convergence Bounds) Assume the base kernel satisfies $W_1(K^n(\delta_x), \nu) \leq \rho^n W_1(\delta_x, \nu)$. Substituting this into the result from Property 3:

$$W_1 \left( K_F^n \left( \delta_z \right), \nu_F \right) \leq L_F \rho^n W_1 \left( \delta_{x_0}, \nu \right). \tag{83}$$

To obtain the second bound, we need to relate $W_1(\delta_{x_0}, \nu)$ back to $\nu_F$. Note that $x_0 = F^{-1}(z)$ and $\nu = F^{-1}\#\nu_F$. Applying the Lipschitz bound for the inverse map $F^{-1}$ (analogous to Property 2):

$$W_1(\delta_{x_0}, \nu) = W_1(F^{-1}\#\delta_z, F^{-1}\#\nu_F) \leq L_{F^{-1}} W_1(\delta_z, \nu_F). \tag{84}$$

Combining these inequalities yields:

$$W_1 \left( K_F^n \left( \delta_z \right), \nu_F \right) \leq L_F L_{F^{-1}} \rho^n W_1(\delta_z, \nu_F). \tag{85}$$

$\square$

**Corollary D.6** (Markov Kernels and Linear Interpolants)**.** *Let $\mathcal{X} = \mathbb{R}^D$. Let $K$ be a Markov transition kernel invariant with respect to a probability measure $\nu$. Let $F_{t|z}(x) = (1-t)z + tx$ for $t \in (0, 1]$ and $z \in \mathcal{X}$.*

*Define the pushforward kernel $K_{t|z}$ by:*

$$K_{t|z}(\hat{x}, \hat{A}) = K \left( F_{t|z}^{-1}(\hat{x}), F_{t|z}^{-1}(\hat{A}) \right). \tag{86}$$

*The following properties hold:*

1. *(Invariant measure) The pushforward kernel $K_{t|z}$ is invariant with respect to $\nu_{t|z}$.*

2. *(Algebraic Iteration) For any measure $\mu$ and $n \geq 1$:*

$$K_{t|z}^n (\mu) = F_{t|z}\# \left( K^n \left( F_{t|z}^{-1}\#\mu \right) \right). \tag{87}$$

3. *(Wasserstein Equality) For any two measures $\mu_1, \mu_2$:*

$$W_1 \left( F_{t|z}\#\mu_1, F_{t|z}\#\mu_2 \right) = t \, W_1 \left( \mu_1, \mu_2 \right). \tag{88}$$

4. *(Kernel Wasserstein Equality) Since $F_{t|z}^{-1}(z) = z$, the bound becomes an equality:*

$$W_1 \left( K_{t|z}^n \left( \delta_z \right), \nu_{t|z} \right) = t \, W_1 \left( K^n \left( \delta_z \right), \nu \right). \tag{89}$$

5. *(Convergence Bounds) If $W_1(K^n(\delta_z), \nu) \leq \rho^n W_1(\delta_z, \nu)$, then:*

$$W_1 \left( K_{t|z}^n \left( \delta_z \right), \nu_{t|z} \right) \leq t \, \rho^n \, W_1 \left( \delta_z, \nu \right), \tag{90}$$

$$W_1 \left( K_{t|z}^n \left( \delta_z \right), \nu_{t|z} \right) \leq \rho^n \, W_1 \left( \delta_z, \nu_{t|z} \right). \tag{91}$$

*Proof.* Results are followed by applying Proposition 1 to the specific linear bijection $F_{t|z}(x) = (1-t)z + tx$. Note that:

- (Lipschitz Constants) Since $F_{t|z}$ is a homothety with scaling factor $t$, we have $\|F_{t|z}(x) - F_{t|z}(y)\| = t\|x - y\|$ and $\|F_{t|z}^{-1}(x) - F_{t|z}^{-1}(y)\| = t^{-1}\|x - y\|$. Thus, $L_F = t$ and $L_{F^{-1}} = t^{-1}$.

- (Fixed Point) Direct evaluation shows $F_{t|z}(z) = z$. Thus, the preimage of the starting point is $x_0 = F_{t|z}^{-1}(z) = z$.

Properties 1 and 2 are independent of the specific properties of the map. Property 5 follows by plugging in the values of $L_F$ and $L_{F^{-1}}$.

3. (Wasserstein Scaling) Proposition 1 (Property 3) provides the upper bound:
$$W_1\left(F_{t|z}\#\mu_1, F_{t|z}\#\mu_2\right) \leq tW_1\left(\mu_1, \mu_2\right). \tag{92}$$

To prove equality, we apply the same general bound to the *inverse* map $F_{t|z}^{-1}$ acting on the measures $\nu_1 = F_{t|z}\#\mu_1$ and $\nu_2 = F_{t|z}\#\mu_2$:
$$W_1\left(\mu_1, \mu_2\right) = W_1\left(F_{t|z}^{-1}\#\nu_1, F_{t|z}^{-1}\#\nu_2\right) \leq t^{-1}W_1\left(\nu_1, \nu_2\right). \tag{93}$$

Multiplying by $t$ yields $tW_1\left(\mu_1, \mu_2\right) \leq W_1\left(F_{t|z}\#\mu_1, F_{t|z}\#\mu_2\right)$. Combining the upper and lower bounds confirms the equality:
$$W_1\left(F_{t|z}\#\mu_1, F_{t|z}\#\mu_2\right) = tW_1\left(\mu_1, \mu_2\right). \tag{94}$$

4. (Kernel Wasserstein Relation) We start with the result from the previous property using $\mu = \delta_z$:
$$K_{t|z}^n(\delta_z) = F_{t|z}\#\left(K^n(F_{t|z}^{-1}\#\delta_z)\right). \tag{95}$$

Since $z$ is a fixed point ($F_{t|z}^{-1}(z) = z$), this simplifies to $F_{t|z}\#(K^n(\delta_z))$. Additionally, $\nu_{t|z} = F_{t|z}\#\nu$. Applying the exact scaling law derived in Item 2 with $\mu_1 = K^n(\delta_z)$ and $\mu_2 = \nu$:
$$W_1\left(K_{t|z}^n(\delta_z), \nu_{t|z}\right) = W_1\left(F_{t|z}\#K^n(\delta_z), F_{t|z}\#\nu\right) = tW_1\left(K^n(\delta_z), \nu\right). \tag{96}$$

$\square$

# E. Sampling Tasks

In this section, we provide more details about the tasks used in our experimental section. A summary of these tasks can be found in Tab. 1.

## E.1. Gaussian mixture (GM)

This task is inspired by the 2-dimensional Gaussian mixture target first used by Midgley et al. (2023). We consider four target distributions of increasing complexity and dimensionality: GM-2, GMNU-2, GM-16, and GMNU-16. All targets are Gaussian mixture models; the first two are illustrated in Fig. 8. The notation GMNU denotes non-uniform mixture weights, meaning that different components carry unequal probability mass, while the suffix indicates the dimensionality ($D = 2$ or $D = 16$). As dimensionality increases, generating high-quality samples becomes more challenging. Similarly, the non-uniform setting is expected to be harder, since a successful sampler must accurately capture the relative mass of each mode.

## E.2. Lennard-Jones (LJ)

We consider the Lennard-Jones (LJ) potential (Jones, 1924), a standard model for solid-state systems and rare-gas clusters. The potential energy for a system of $N$ particles is defined as:
$$E(x_1, \ldots, x_N) = \sum_{1 \leq i < j \leq N} 4\epsilon\left[\left(\frac{\sigma}{r_{ij}}\right)^{12} - \left(\frac{\sigma}{r_{ij}}\right)^6\right], \tag{97}$$

where $x_n \in \mathbb{R}^3$, $r_{ij} = \|x_i - x_j\|_2$ is the Euclidean distance between particles $i$ and $j$, and $\epsilon > 0$ and $\sigma > 0$ are physical constants. Following prior work (Klein et al., 2023), we consider two Boltzmann distributions induced by this potential: LJ-13 and LJ-55, corresponding to systems with 13 and 55 particles, and dimensionalities $D = 39$ and $D = 155$, respectively. To assess the quality of the generated samples, we use the test set provided by (Klein et al., 2023) as ground-truth samples.

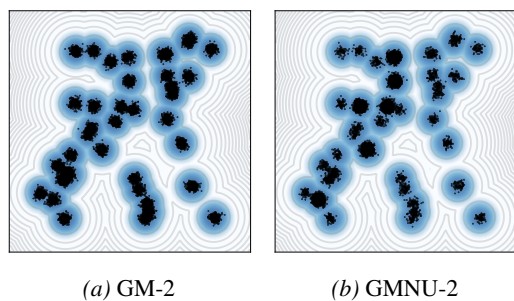

*(a)* GM-2          *(b)* GMNU-2

*Figure 8.* GM task in two dimensions.

### E.3. Alanine Dipeptide (ALDP)

We consider the alanine dipeptide molecule in vacuum at $T = 300\,\mathrm{K}$, a standard benchmark for evaluating sampling methods (Smith, 1999). The system comprises 22 atoms, corresponding to a $D = 66$-dimensional configuration space, and the target is the Boltzmann distribution induced by the molecular force field. Since potential evaluations are computationally expensive, this setting highlights the need for samplers that achieve high sample quality with few density evaluations. Ground truth samples are generated using OpenMM molecular dynamics simulations following the setup of Midgley et al. (2023).

**Ramachandran Histograms.** To evaluate the quality of the generated samples, we project the 66-dimensional configuration space onto the two principal dihedral angles, $\phi$ and $\psi$ (Midgley et al., 2023). These projection is done using the `mdtraj` package (McGibbon et al., 2015). This results in a 2D histogram, known as a Ramachandran plot, which provides a visually intuitive and physically meaningful representation of the molecule's conformational states. Following previous work (Noé et al., 2019; Midgley et al., 2023; Rissanen et al., 2025), we use these plots to identify whether each sampler effectively navigates these barriers to achieve a global exploration of the conformational space compared to the MD ground truth.

**Chirality.** Alanine dipeptide is a chiral molecule, meaning it can exist in two forms: the L-form and the D-form (Midgley et al., 2023). While the D-form is found in synthetically created compounds, the L-form appears almost exclusively in nature. Consequently, the literature focuses predominantly on the L-form (Smith, 1999). To restrict our experiments to this biologically relevant state, we filter out D-forms by adding a penalization term to the potential energy surface. This term assigns a high energy penalty to D-configurations while remaining zero for L-configurations. Formally, this term is defined as the signed volume, $\Omega$, of the parallelepiped spanned by the vectors connecting the central alpha-carbon ($C_\alpha$) to its neighbors. This is defined by the scalar triple product:

$$\Omega = (\mathbf{r}_N - \mathbf{r}_{C_\alpha}) \cdot \left[ (\mathbf{r}_C - \mathbf{r}_{C_\alpha}) \times (\mathbf{r}_{C_\beta} - \mathbf{r}_{C_\alpha}) \right], \tag{98}$$

where $\mathbf{r}_X$ represents the position vector of atom $X \in \{N, C, C_\beta, C_\alpha\}$. The sign of $\Omega$ determines the chirality, allowing us to penalize the system when it transitions into the D-region.

### E.4. Bayesian Neural Network (BNN)

We consider the problem of sampling from the posterior distribution over the parameters of a Bayesian neural network, which is known to be complex and highly multimodal (Neal, 2012; Izmailov et al., 2021). Such samples are required to accurately compute posterior expectations and to quantify predictive uncertainty. Evaluating the target density involves a forward pass through the network, while computing the score of the log density requires a backward pass, both of which scale with the number of parameters. This makes it crucial to minimize the number of density evaluations needed to obtain high-quality samples.

## F. Evaluation Metrics

To evaluate performance, we consider a diverse set of metrics: Maximum Mean Discrepancy (MMD), Total Variation distance, the Wasserstein-2 distance ($W_2$), and the relative Mean Absolute Error (MAE). In addition, we report two task-specific metrics: the Ramachandran KL divergence for the ALDP task, and the Test Negative Log Likelihood (Test NLL) for the BNN task. Aggregate performance is quantified by the Mean Hypervolume Ratio (HVR) (Zitzler et al., 2003), computed

on normalized Pareto fronts to ensure comparability across objectives of varying scales. Next, we describe these metrics.

**Maximum Mean Discrepancy (MMD).** We use the definition of MMD given by Gretton et al. (2012). We compute MMD on the histograms of negative log-densities of the samples. Given two sets of samples $X = \{x_1, \ldots, x_n\}$ and $Y = \{y_1, \ldots, y_m\}$, the squared MMD estimate is defined as:

$$\widehat{\text{MMD}}^2(X, Y) = \frac{1}{n(n-1)} \sum_{i \neq j} k(x_i, x_j) - \frac{2}{nm} \sum_{i,j} k(x_i, y_j) + \frac{1}{m(m-1)} \sum_{i \neq j} k(y_i, y_j) \quad (99)$$

where $k(\cdot, \cdot)$ is a positive definite kernel. In this work, we use the Gaussian kernel $k(x, y) = \exp\left(-\frac{\|x-y\|^2}{2\sigma^2}\right)$.

**Total Variation (TV).** We approximate the Total Variation distance using the discretized histograms of the negative log-densities. Let $H_P$ and $H_Q$ be the normalized histogram vectors (probability mass functions) for the true and generated data respectively, where $H(i)$ represents the probability mass in the $i$-th bin. The TV distance is computed as:

$$\text{TV}(H_P, H_Q) = \frac{1}{2} \sum_i |H_P(i) - H_Q(i)| \quad (100)$$

**Wasserstein-2 Distance ($W_2$).** The Wasserstein-2 distance measures the cost of transporting the generated distribution to the target distribution. In practice, we compute $W_2$ using the Python Optimal Transport package (Flamary et al., 2021).

For the Lennard-Jones (LJ) task, the metric $d(x, y)$ must account for rotational and translational symmetries. Following previous work (Rissanen et al., 2025), we incorporate the Kabsch algorithm to ensure equivariance. The distance between two configurations $x$ and $y$ is defined as the root-mean-square deviation (RMSD) after optimal superposition:

$$d_{\text{Kabsch}}(x, y) = \min_{R \in SO(3), t \in \mathbb{R}^3} \|Rx + t - y\|_2 \quad (101)$$

**Relative Mean Absolute Error (MAE).** The relative MAE evaluates the accuracy of the estimated expectation of a scalar observable $f(x)$. We use the same quadratic observable as in Midgley et al. (2023); Chen et al. (2024). The relative error is given by:

$$\text{RelMAE} = \frac{|\mathbb{E}_{x \sim \hat{\nu}}[f(x)] - \mathbb{E}_{x \sim \nu}[f(x)]|}{|\mathbb{E}_{x \sim \nu}[f(x)]|} \quad (102)$$

**Ramachandran Kullback–Leibler Divergence (Ramachandran KL).** For the ALDP task, we assess the structural fidelity by comparing the marginal distributions of the dihedral angles $\phi$ and $\psi$. We compute the Kullback–Leibler (KL) divergence between discretized Ramachandran histograms of the ground-truth and generated samples. See App. E for more details about the Ramachandran histograms.

**Ramachandran Root Mean Squared Error (Ramachandran RMSE).** To complement the KL divergence, we evaluate the thermodynamic landscapes through the Free Energy Surface (FES) obtained from the Ramachandran histograms. We quantify the discrepancy between the ground-truth and generated FES using the Root Mean Square Error (RMSE). See App. E for details about the Ramachandran histograms.

**Test Negative Log Likelihood (Test NLL).** For the BNN task, ground-truth samples from the posterior are unavailable, making the previous metrics inapplicable. Instead, we evaluate models using the average Negative Log Likelihood (NLL) on a held-out test set. For each posterior sample (corresponding to a set of network weights), we compute the test NLL and then report the average across samples.

**Hypervolume Ratio (HVR).** To quantitatively assess the quality of the Pareto fronts across tasks with varying objective scales, we employ the Hypervolume (HV) indicator (Zitzler et al., 2003). The HV indicator measures the volume of the objective space that is dominated by a set of non-dominated solutions, bounded by a reference point. It is widely regarded as the gold standard in multi-objective optimization because it is strictly monotonic with respect to Pareto dominance. This means that a set of solutions that improves upon another in terms of convergence or diversity will strictly yield a higher hypervolume.

Since the metrics in our set of tasks operate on different scales, raw hypervolume scores cannot be aggregated directly. To enable a fair cross-task comparison, we compute the Mean Hypervolume Ratio (HVR) following standard benchmarking

protocols (Zitzler et al., 2003). For each target distribution and evaluation metric, we employ the following procedure to compute the HVR:

1. Normalization: We identify the global minimum and maximum objective values across all methods and all evaluations. The objective vectors for all methods are then normalized to the unit square $[0, 1]^2$ via linear rescaling.

2. Reference Front Construction: We construct a *best known* Pareto front for each task by pooling the solutions from all methods and filtering for the non-dominated set. The reference hypervolume, $HV_{\text{ref}}$, is computed based on this combined front using a reference point of $(1.1, 1.1)$ in the normalized space to ensure all boundary solutions are captured.

3. Ratio Calculation: The HVR for a specific method is defined as the ratio of its hypervolume to the reference hypervolume:

$$\text{HVR}(\texttt{method}) = \frac{HV(\texttt{method})}{HV_{\text{ref}}}. \tag{103}$$

An HVR of $1.0$ indicates that a method has successfully recovered the entire best-known Pareto front, while lower values indicate a failure to converge or a lack of diversity in the solution set. To obtain the Mean HVR of a method, we average the HVR corresponding to that method over the selected metrics and targets.

## G. Experimental Setup

In this section, we describe the implementation details and hyperparameter configuration of the methods used in our experiments. We also explain how to obtain the Pareto fronts depicted in Fig. 6 and App. H.

**Implementation.** All methods and target distributions are implemented in PyTorch, closely following their original formulations. The code used in our experiments has been uploaded as supplementary material.

For NRPT, we extend the implementation by Rissanen et al. (2025), which is available at `https://github.com/cambridge-mlg/Progressive-Tempering-Sampler-with-Diffusion`. For OASMC, we adapt to PyTorch the pseudo-code provided by Syed et al. (2024). For DiGS, we rely on the implementation available at the official repository: `https://github.com/Wenlin-Chen/DiGS`. The implementations of MALA and HMC are standard, and we therefore use conventional versions of these algorithms. In all cases, these implementations have been refined to maximize performance while minimizing density evaluations. The implementation of CDS closely follows Alg. 1.

Finally, all samplers are implemented to prioritize computational efficiency by minimizing target density evaluations. To achieve this, each sampler caches its current state, comprising the sample coordinates $x$, the log unnormalized density $\log \pi(x)$, and the score $\nabla \log \pi(x)$. This caching mechanism prevents redundant gradient and density computations during internal steps.

**Methods, Hyperparameters and Configuration.** In this work, we consider the following methods: Non-Reversible PT (NRPT) (Syed et al., 2022), Optimized Annealed SMC (OASMC) (Syed et al., 2024), Diffusive Gibbs Sampling (DiGS) (Chen et al., 2024), Metropolis–Adjusted Langevin Algorithm (MALA), Hamiltonian Monte Carlo (HMC), No-U-Turn Sampler (NUTS, Hoffman et al. (2014)), Stein Variational Gradient Descent (SVGD, Liu & Wang (2016)), and the Metropolis Adjusted Microcanonical Sampler (MAMS, Robnik et al. (2025)).

To ensure a fair evaluation, we explore multiple hyperparameter values for each method and select the optimal configurations via grid search.

Several settings are shared across the evaluated methods. For the exploration (or denoising) kernels, all methods utilize the Metropolis-Adjusted Langevin Algorithm (MALA) across all tasks, with the exception of the ALDP task, which utilizes Hamiltonian Monte Carlo (HMC). To ensure numerical stability, we initialize a task-specific base step size that is consistent across all samplers: 0.1 for the GM task, 0.0001 for the LJ task, 0.000001 for the ALDP task, and 0.00001 for the BNN task. From this base initialization, all methods employ an adaptive step size: following each step, the size is updated to target theoretically optimal acceptance rates of $0.574$ for MALA and $0.651$ for HMC (Roberts & Rosenthal, 1998; Beskos et al., 2013). Furthermore, for methods requiring a reference distribution (NRPT and OASMC), we use a flat distribution (i.e., a uniform distribution with infinite support) in the GM, LJ, and ALDP tasks, and the prior distribution in the BNN task.

The method-specific hyperparameters are detailed below:

- **CDS:** The hyperparameters include the first-stage sampler configurations, the number of integration steps, the corrector kernel and steps, the initial time $t_0$, and the noise schedule $\sigma_t$. We employ NRPT for the first stage, fixing its hyperparameters to near-optimal values obtained for the NRPT baseline. We note that these values were not explicitly jointly optimized for CDS, where the first stage targets a distribution closer to the reference than the standalone NRPT baseline (and thus we expect them to differ). For the integration phase, we consider $\{10, 100, 1000\}$ for the integration steps and tune $t_0$ independently for each task ($t_0 = 0.01$ for GM, $t_0 \in \{0.2, 0.3\}$ for LJ, $t_0 \in \{0.1, 0.2\}$ for ALDP, $t_0 = 0.1$ for BNN). We utilize MALA as the corrector kernel, applying either 0 (no correction) or 1 corrector step. Finally, we employ a constant noise schedule, fixing $\sigma_t$ to the same task-specific base step size shared across the other evaluated methods.

- **NRPT:** The hyperparameters include the number of replicas and the annealing schedule. We select the number of replicas from the set $\{3, 5, 10\}$. For the annealing schedule, we adopt a geometric schedule and tune the initial value, $\beta_{\min}$, per task ($\beta_{\min} \in \{0.001, 0.01\}$ for GM, $\beta_{\min} \in \{0.7, 0.8\}$ for LJ, $\beta_{\min} \in \{0.3, 0.5\}$ for ALDP, $\beta_{\min} \in \{0.7, 0.8\}$ for BNN). Additionally, we consider the schedule optimization procedure proposed by (Syed et al., 2022).

- **OASMC:** Hyperparameters include the number of particles, the resampling threshold, the resampling mechanism, and the annealing schedule. We consider $\{10, 100, 1000\}$ particles for all tasks. For the resampling mechanism, we used both multinomial and systematic resampling, and observed that the latter provides a stronger empirical performance (Smith, 2013). Resampling events are triggered whenever the effective sample size (ESS) proportion falls below a specified threshold, evaluating values in $\{0.5, 1.0\}$. Similar to NRPT, we utilize a geometric annealing schedule, tune $\beta_{\min}$ per task, and apply the schedule optimization procedure introduced by (Syed et al., 2024).

- **DiGS:** The key hyperparameters are the noise schedule, the number of denoising steps, and the number of noise levels. The noise schedule follows the one used in the original paper (Chen et al., 2024). It is determined by the parameters $\alpha_{\min}$ and $\alpha_{\max}$, for which we consider $\{0.1, 0.4\}$ and $\{0.6, 0.9\}$, respectively. We consider $\{1, 4\}$ for the number of denoising steps and $\{1, 5\}$ for the number of noise levels.

- **HMC:** In addition to the shared adaptive step size, we tune the number of leapfrog steps, considering the set $\{3, 5\}$ across all tasks.

- **MALA:** The sole hyperparameter is the step size, which relies entirely on the shared adaptive procedure described above.

- **NUTS:** In addition to the shared adaptive step size, we tune the maximum tree depth (which bounds the trajectory length); we consider the set $\{3, 5, 8\}$ for all tasks.

- **MAMS:** Similar to HMC, we tune the number of integration steps, considering the set $\{3, 5\}$ across all tasks.

- **SVGD:** The key hyperparameters include the step size (learning rate) for the particle updates and the kernel configuration. We adopt the shared task-specific base step size used across the other evaluated methods, and employ a Radial Basis Function (RBF) kernel with its bandwidth determined via the median heuristic.

**Initialization and Sample Generation.** To ensure a fair comparison, all samplers share a common initialization scheme. First, we identify one mode of each target density, $x_\pi^* = \arg\max_x \log \pi(x)$, using $1,000$ iterations of standard gradient descent to ensure convergence. All samplers are then initialized at $x_\pi^*$. For CDS, which requires initialization from a reference distribution $z \sim \nu_{\text{ref}}$, we define $\nu_{\text{ref}} = \mathcal{N}(x_\pi^*, \tau^2 I)$ and set the starting point to the mean, $z = x_\pi^*$. We fix $\tau = 1.0$ across all target densities.

For each method, the total sample count is set to $10^4$ for the GM and LJ tasks, $10^5$ for the ALDP task, and $10^3$ for the BNN task. To minimize sample autocorrelation, we employ a fully parallelized scheme where each sampler runs as many chains as the number of required samples. This is implemented efficiently in PyTorch using vectorized operations, generating a tensor of shape (num_samples, $D$).

**Pareto Fronts.** To construct the Pareto fronts presented in this paper, for each dataset, we fix a set of computational budgets, measured in terms of the number of density evaluations. For each budget and each method, we run all hyperparameter configurations for a number of iterations chosen to match the prescribed budget. Each configuration is repeated three times to obtain uncertainty estimates.

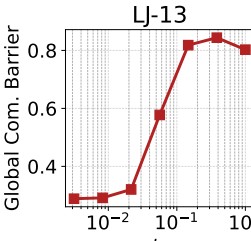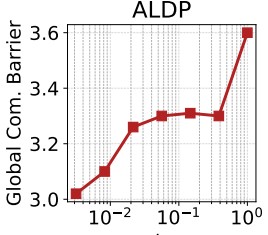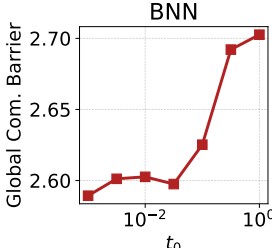

*Figure 9.* **Global Communication Barrier (GCB) as a function of** $t_0$. Across LJ-13, ALDP, and BNN, decreasing $t_0$ initially improves communication efficiency (lower GCB).

Pareto fronts are then constructed using a bootstrap procedure adapted from (Grunert da Fonseca et al., 2001). For each method, we perform 50 bootstrap iterations; in each iteration, we resample the experimental replicates to estimate the mean computational cost and performance. We then identify the non-dominated configurations and represent the resulting frontier as a monotonic step function. Finally, we aggregate the bootstrapped frontiers to compute the median Pareto front, together with 5th–95th percentile confidence bands.

## H. Additional Experiments and Results.

### H.1. Communication Efficiency Analysis.

We study the communication efficiency of PT when targeting the conditional distribution $\nu_{t|z}$. Recall that the first stage of CDS samples from $\nu_{t_0|z}$ for a fixed $t_0 > 0$. As $t \to 0$, $\nu_{t|z}$ concentrates to a Dirac mass at $z$ (a sample from the reference distribution), suggesting that communication should improve for small $t$. To test this hypothesis, we run NRPT targeting $\nu_{t_0|z}$ across different values of $t_0$, keeping all hyperparameters fixed except for $t_0$.

Two factors influence PT communication efficiency: the annealing schedule and the number of replicas. We optimize the annealing schedule for each run, and set the number of replicas to the ceiling of the GCB estimated from a pilot run, following (Syed et al., 2022). These choices effectively control for their impact, isolating the effect of $t_0$ on communication efficiency.

**Round Trips (RTs).** RTs is the number of times a replica traverses between the reference and target distributions. It is usually used as the main measure to assess PT efficiency, as higher RTs indicate better mixing (Syed et al., 2022). As shown in Fig. 4, decreasing $t_0$ from 1.0 generally increases RTs across all tasks, confirming that sampling $\nu_{t_0|z}$ is more efficient than sampling $\nu$ directly. However, efficiency starts to drop as $t_0 \to 0$, where the distribution becomes excessively peaked, reducing replica overlap. As depicted in Fig. 3, this suggests an optimal range for $t_0$: small enough so that the target overlaps with the reference, but high enough to avoid singularities.

**Global Communication Barrier.** In addition to Round Trips, we evaluate communication efficiency using the GCB (Syed et al., 2022). The GCB aggregates the rejection probabilities of swap proposals between adjacent replicas, providing a global measure of the difficulty of the annealing path. Lower GCB values indicate easier communication and higher overall efficiency. The GCB results are reported in Fig. 9. Consistent with the RT analysis, we observe that decreasing $t$ generally leads to improved communication, as reflected by a decreasing GCB in the LJ-55, ALDP, and BNN tasks. This further supports the claim that the conditional distributions $\nu_{t|z}$ are easier to sample than the original target.

**Sample Quality.** Sample quality improves as $t_0$ decreases from 1.0, reflecting the enhanced communication efficiency of PT in this regime. However, this trend reverses as $t_0 \to 0$: the target distribution becomes increasingly concentrated, reducing overlap between replicas and degrading mixing. As a result, sample quality deteriorates for very small $t_0$, indicating the existence of an optimal intermediate range that balances overlap and numerical stability.

### H.2. Initializing with a Different Sampler

We study the impact of the sampler used in Stage 1 of CDS. The objective of this stage is to draw samples from $\nu_{t_0|z}$ for a fixed $t_0 > 0$. In principle, any sampling procedure could be employed for this purpose. However, since $\nu_{t_0|z}$ remains multimodal, we expect *local* MCMC methods (e.g., MALA or HMC) to exhibit poor mixing. In contrast, annealing-based

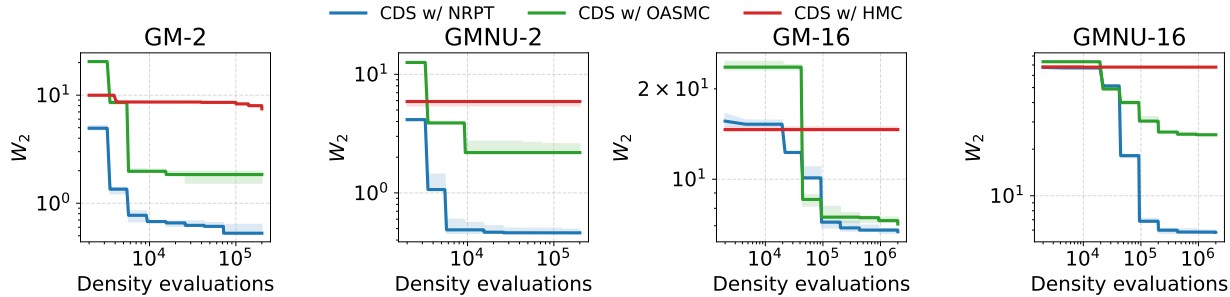

*Figure 10.* **Effect of the Stage 1 sampler on CDS performance.** We compare NRPT, OASMC, and HMC while keeping Stage 2 (SDE integration) fixed. NRPT consistently outperforms OASMC, whereas HMC struggles to explore the multimodal distribution and exhibits significantly degraded performance.

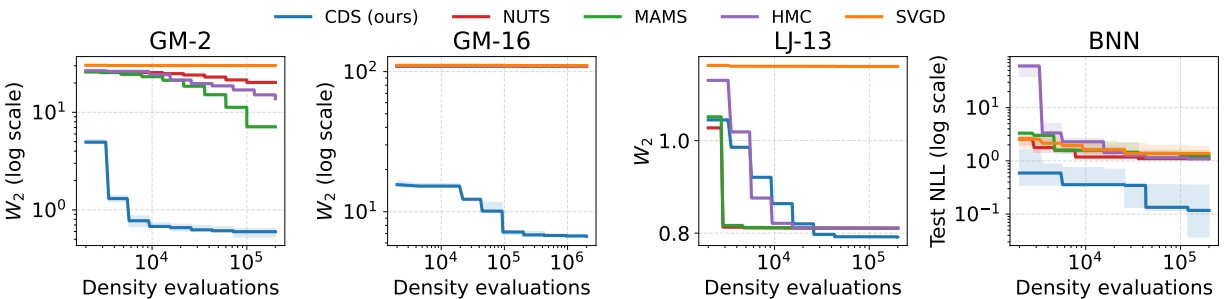

*Figure 11.* **Comparison of CDS against additional baselines (NUTS, SVGD, MAMS).** CDS consistently outperforms all methods in GM and BNN, while in LJ, MAMS and NUTS perform best in the low-budget regime but are surpassed by CDS as the number of density evaluations increases.

methods (such as PT and SMC), which progressively bridge the reference and target distributions, are better suited to this setting, particularly because $\nu_{t_0|z}$ approaches the reference distribution as $t_0 \to 0$.

We compare CDS performance under different choices of sampler in Stage 1. Specifically, we consider NRPT, OASMC, and HMC, all tuned as described in App. G, while keeping the parameters of Stage 2 (SDE integration) fixed. The results, shown in Fig. 10, indicate that although OASMC is a competitive alternative, it is consistently outperformed by NRPT. In contrast, HMC performs poorly: it becomes trapped in local modes and fails to adequately explore the target distribution.

### H.3. Additional Baselines

We compare the proposed CDS against additional baselines consisting of variants of the local samplers (MALA and HMC) considered in the main text. Specifically, we include the No-U-Turn Sampler (NUTS, Hoffman et al. (2014)), Stein Variational Gradient Descent (SVGD, Liu & Wang (2016)), and the Metropolis Adjusted Microcanonical Sampler (MAMS, Robnik et al. (2025)). Results are reported in Fig. 11. As NUTS and MAMS are extensions of HMC, we also include HMC for reference.

In the GM task, CDS consistently outperforms all baselines. The competing methods exhibit behavior similar to HMC and MALA, becoming trapped in local modes and failing to mix effectively. In the LJ task, MAMS and NUTS achieve near-optimal performance with fewer than $10^4$ density evaluations, outperforming both HMC and CDS in this low-budget regime. This is expected, as local samplers such as MALA and HMC are sufficient to accurately capture the target distribution in this setting. However, as the computational budget increases, CDS surpasses all methods and achieves the best overall performance. Finally, in the BNN task, CDS again delivers the strongest results. While the additional baselines improve upon HMC, they remain unable to match the performance of CDS.

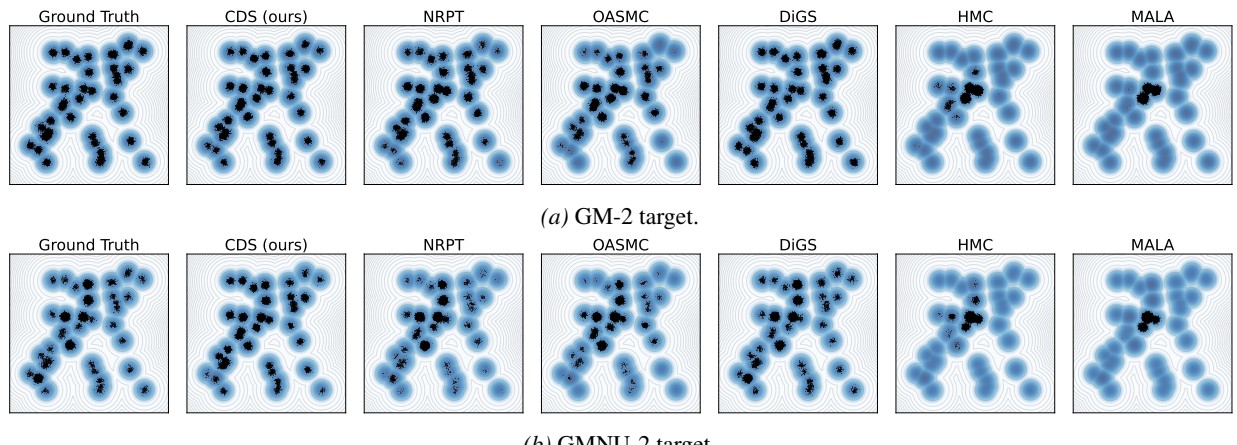

*(a)* GM-2 target.

*(b)* GMNU-2 target.

*Figure 12.* **Comparison of ground truth and generated samples for the Gaussian Mixture (GM) task.** Each method uses a fixed budget of $2 \cdot 10^3$ density evaluations.

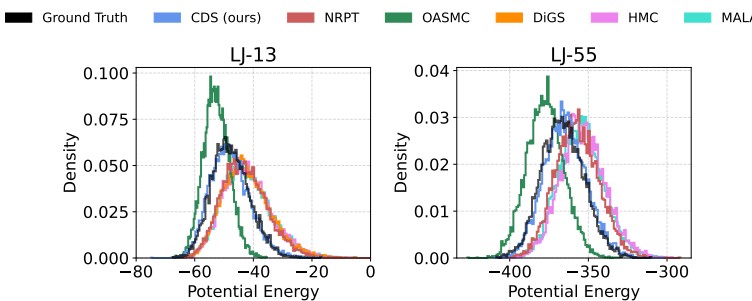

*Figure 13.* **Comparison of Lennard-Jones (LJ) potential energy histograms.** All samplers use a fixed budget of $2 \cdot 10^4$ and $2 \cdot 10^5$ density evaluations for the LJ-13 and LJ-55 targets, respectively. DiGS is omitted from the LJ55 plot as it produces a degenerate histogram.

## H.4. Qualitative Analysis

In this section, we analyze the samples generated by each method across the different tasks. Our goal is to determine how quantitative performance differences translate into qualitative sample fidelity.

**GM Task.** For the GM task, we focus on the GM-2 and GMNU-2 targets, as their two-dimensional nature allows for direct visualization. As shown in Fig. 12, when methods are restricted to a limited budget of density evaluations, only the proposed CDS and DiGS successfully recover all modes. This aligns with the Pareto front results in Fig. 17, where CDS and DiGS demonstrated superior performance.

**LJ Task.** Energy histograms for the LJ systems are displayed in Fig. 13. We observe that CDS most accurately reproduces the target energy distribution, followed by NRPT and OASMC. Notably, for the LJ-55 target, only CDS and NRPT are able to accurately approximate the target distribution.

**ALDP Task.** Ramachandran histograms for the ALDP task are presented in Fig. 2. The ground truth samples reveal two distinct metastable states separated by a high-energy barrier. Both CDS (ours) and NRPT successfully recover these modes with accurate density allocations; however, NRPT slightly outperforms CDS in this specific instance. Conversely, OASMC and DiGS exhibit artifacts in high-energy transition regions, while local samplers (MALA and HMC) suffer from mode collapse, failing to traverse the energy barrier.

## H.5. Ablation study

We analyze the impact of key hyperparameters on CDS performance: the computational budget allocation between phases, the number of integration steps, the use of corrector steps, and the choice of noise schedule.

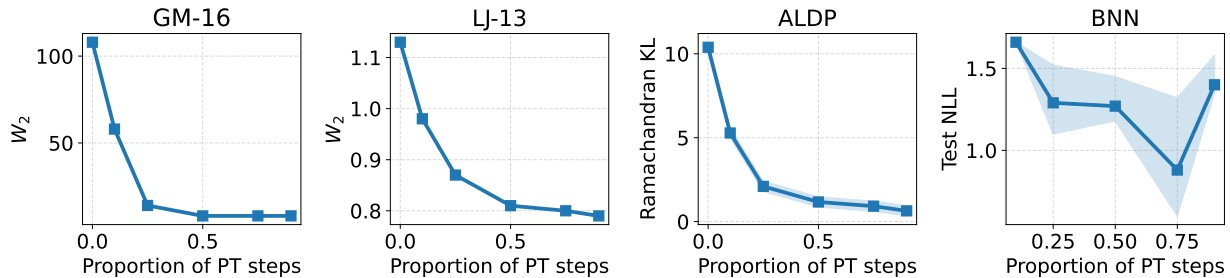

*Figure 14.* **Impact of steps allocation on CDS performance**. We evaluate the trade-off between Parallel Tempering (PT) and SDE integration by varying the proportion of total steps dedicated to the PT phase.

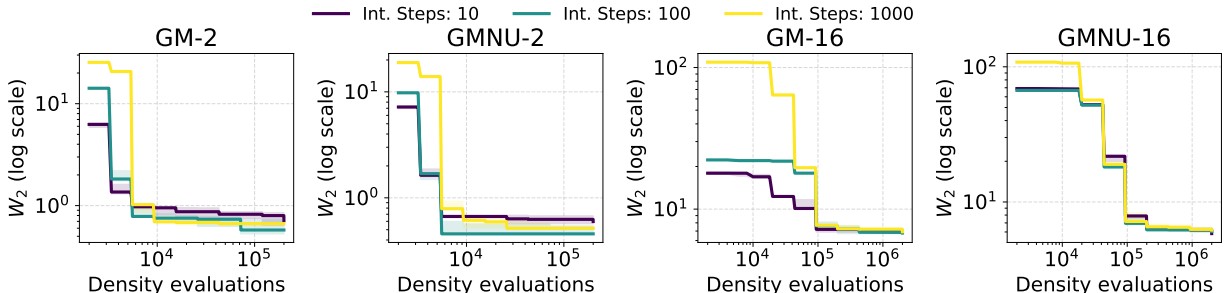

*Figure 15.* **Impact of integration steps on CDS performance.** Under a limited evaluation budget, fewer integration steps are preferable as they allow more resources for exploration in the first stage. In contrast, with a larger budget, increasing the number of integration steps improves the accuracy of the transport to the target distribution.

**Computational Budget Allocation Between Phases.** We study the balance between the two stages of CDS to understand how to allocate computational resources effectively. To this end, we introduce a hyperparameter $\rho$, which controls the fraction of steps assigned to Stage 1. Given a total computational budget of $N$ steps, we define:

$$N_{\text{Stage1}} = \rho N, \qquad N_{\text{Stage2}} = (1 - \rho)N. \tag{104}$$

Thus, $\rho$ determines the relative emphasis between the two stages of CDS, enabling a controlled analysis of their trade-off.

We fix all other configurations and evaluate performance as a function of $\rho$ across tasks, see Fig. 14. As expected, $\rho = 0$ (pure SDE integration) performs poorly, confirming that starting the integration at $t = 0$ leads to insufficient exploration of the target distribution. Increasing $\rho$ consistently improves performance by incorporating more PT steps, which enhance global exploration.

For the GM, LJ, and ALDP tasks, performance continues to improve as $\rho$ increases, indicating that only a small number of integration steps are sufficient to transport samples effectively to the target distribution. In contrast, for the BNN task, overly large values of $\rho$ degrade performance, as too few SDE integration steps hinder accurate transport to the target.

**Integration Steps.** From the previous analysis, we observe that allocating more steps to the first stage of CDS is beneficial. Next, we investigate deeper on how to choose the number of integration steps.

As before, we fix all other configurations and evaluate performance as a function of the number of integration steps. Results are shown in Fig. 15, where we report Pareto fronts for different choices of this parameter. We observe that, under a limited density evaluation budget, it is more effective to use fewer integration steps, as this allows more steps to be devoted to exploration in the first stage. In contrast, when the budget is sufficiently large, allocating more integration steps becomes advantageous, improving the accuracy of the transport to the target distribution.

Overall, our results highlight the need for a balanced allocation of computational resources: sufficient effort must be devoted to the PT stage to ensure effective global exploration, while retaining enough integration steps to accurately map samples to the target distribution.

**Diffusion Noise and Corrector Steps.** We study two closely related hyperparameters: the diffusion noise variance ($\sigma_t^2$)

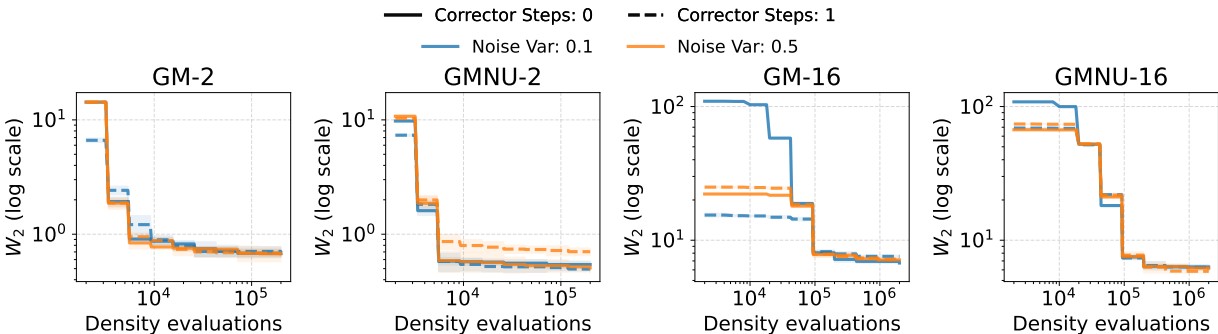

*Figure 16.* **Impact of corrector steps and noise variance on CDS performance.** Corrector steps improve results only at low noise levels, while their impact becomes negligible as the diffusion variance increases.

and the use of corrector steps. Both affect the second stage of the method, in which the interpolation SDE is integrated. Corrector steps were originally introduced to mitigate errors arising from the numerical discretization of the SDE. In turn, the diffusion variance controls the weight of the score term in the SDE, and therefore modulates the strength of the guidance toward the target distribution.

Importantly, the effectiveness of corrector steps depends on the choice of diffusion variance. In particular, smaller variance reduces the influence of the score term, which can lead to samples drifting away from the intermediate distribution at each integration step.

We empirically verify this interaction by evaluating CDS across different configurations of these hyperparameters on the GM task. The results are shown in Fig. 16. We observe that corrector steps improve performance only when the diffusion variance is low. In contrast, for higher noise levels, introducing corrector steps does not yield any noticeable benefit.

## H.6. Additional pareto fronts

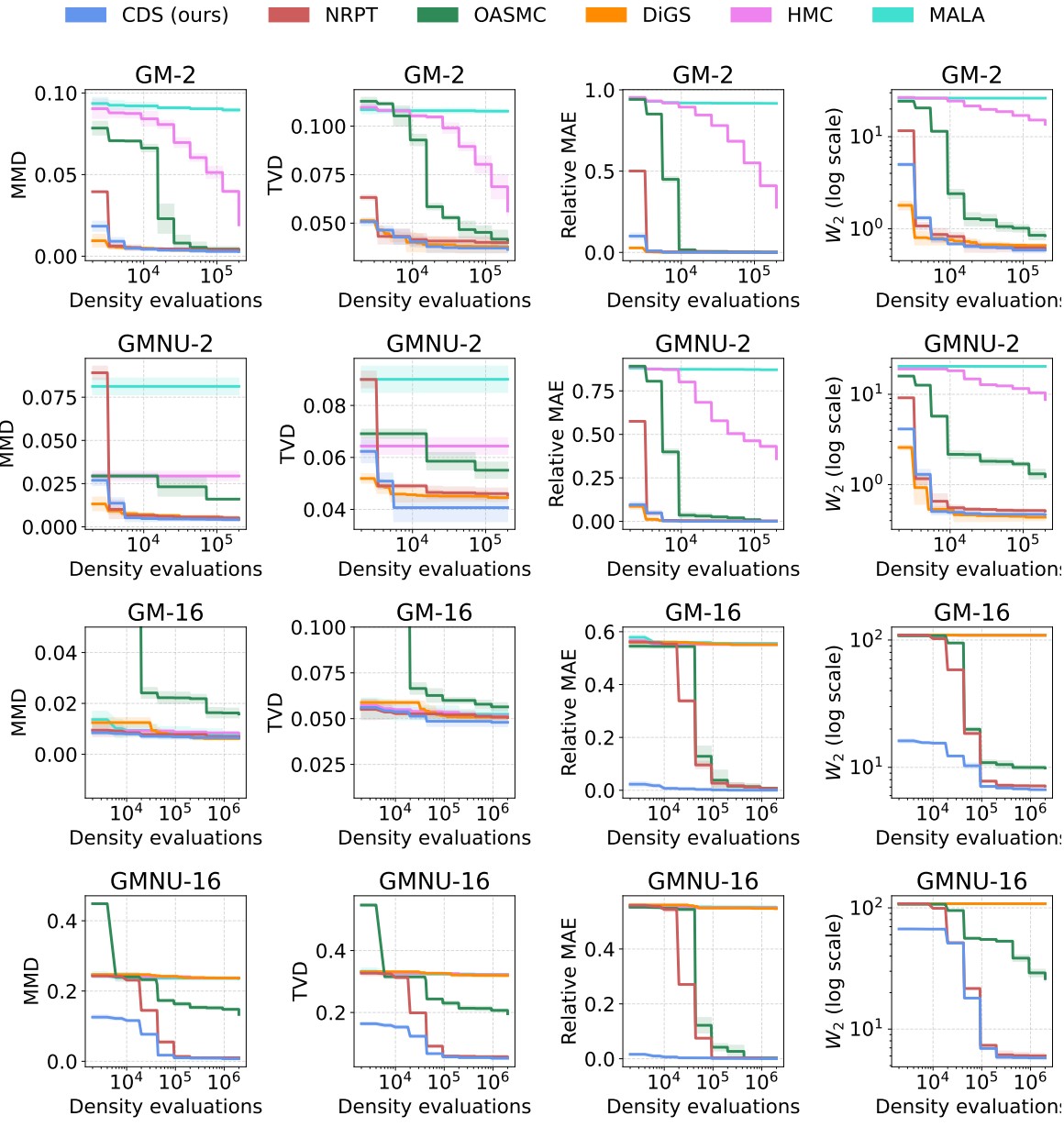

*Figure 17.* **Pareto fronts for the Gaussian Mixture (GM) task across different evaluation metrics**. Evolution of performance across different evaluation criteria (defined in App. F). Each curve represents the optimal trade-off between computational budget and sample quality.

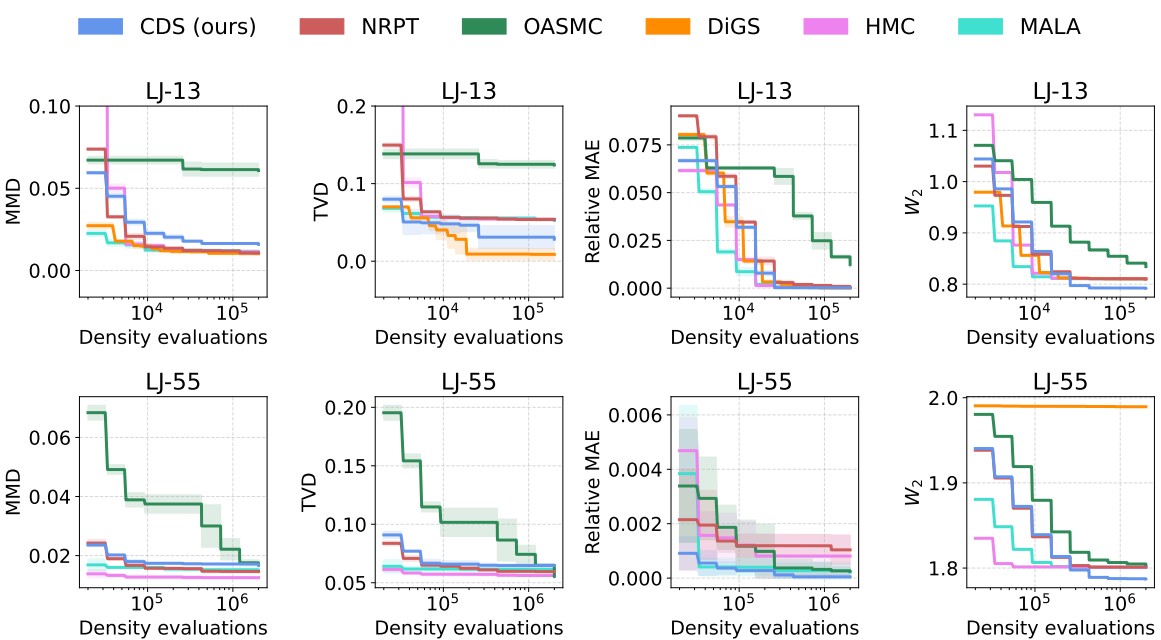

*Figure 18.* **Pareto fronts for the Lennard-Jones (LJ) task across different evaluation metrics**. Evolution of performance across different evaluation criteria (defined in App. F). Each curve represents the optimal trade-off between computational budget and sample quality.

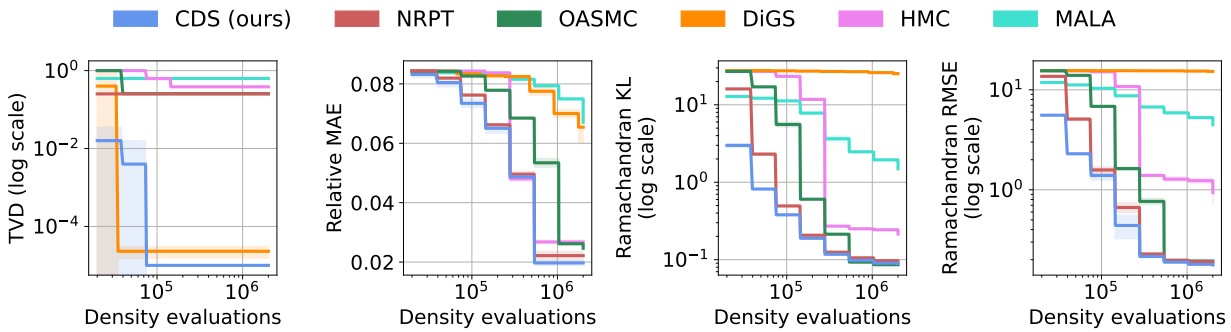

*Figure 19.* **Pareto fronts for the Alanine Dipeptide (ALDP) task across different evaluation metrics**. Evolution of performance across different evaluation criteria (defined in App. F). Each curve represents the optimal trade-off between computational budget and sample quality.

