# OpenReview forum: "Conditional Diffusion Sampling"
_ICML.cc/2026/Conference — ICML 2026 regular_

### Official Review · Reviewer_sovk · 2026-02-24

**Soundness:** 2
**Presentation:** 3
**Significance:** 2
**Originality:** 1
**Overall Recommendation:** 3
**Confidence:** 3

**Summary:**

This paper presents Conditional Interpolants, a framework that leverages the advantages of Parallel Tempering (PT) and diffusion-based methods. Conditional interpolants learn a conditional path, rather than a marginal probability path, to construct bridges between an initial distribution (at a small time t_0 > 0, this is shown to be an advantage from tractability) and a target distribution. The goal is to sample from an unnormalized Gibbs distribution. This is a training-free method, with the score function also in a closed form.

**Compliance With Llm Reviewing Policy:**

Affirmed.

**Key Questions For Authors:**

Can you please clarify how does this method differ from the other frameworks that also use conditional vector fields for tractability and simulation-free training (e.g. Flow Matching and related frameworks)?

Can you clarify that PT in this paper is simply a means to get an initial point $ x_0 $ at time $ t_0 $? I'm not sure what the point of this is.

The second part of the algorithm appears to be a kind of predictor-corrector step using the EM discretization of the Langevin dynamics, can you confirm this?

I am willing to raise my score if I can get an understanding of the novelties of this paper, because it looks like sticking in a conditioning variable is the point, along with parallel tempering.

**Limitations:**

The authors have acknowledged some limitations:
* choice of interpolant is critical for densities with singularities and currently there is no principled way to find them.
* Choice of $ t_0 $ for the PT stage.
* Everything else mentioned in the strengths and weakness section.

**Strengths And Weaknesses:**

I do not understand what this paper is contributing, beyond the fact that the authors have conditioned on a sample from a given reference distribution $ z \sim \nu_{\rm ref}$. Eq (13) is simply the vector field describing the bridge between the state $ x $ and the reference sample $ z $. The parallel tempering method appears to just generate multiple samples from the reference and the usual SDE looks like a conditional diffusion model with conditional vector fields that drive the process to the target distribution. This work says that the score function doesn't need to be learned; it simply needs to be constructed. In the case of linear interpolants, the examples are provided. This method also seems to hold for arbitrary reference measures. Experiments look interesting. Not sure if this is too novel.

Edit: Upon re-reading/literature review, this work appears to be not that incremental to Rissanen et. al's paper "Progressive Tempering Sampler with Diffusion" 2025.

---

> ### Author Rebuttal · Authors · 2026-03-30
>
> We would like to thank the reviewer for their questions. We address them in the following.
>
> **1. Differences from Flow Matching and Simulation-free Training Methods**
>
> As the reviewer points out, "sticking in a conditioning variable" is an important point of our approach. It is the mathematical mechanism that allows us to remove neural network training entirely. We elaborate further on this point below.
>
> **Marginal Paths and Intractability.**
> Standard diffusion and flow matching simulate a stochastic process following a *marginal* probability path to transport samples from a reference distribution to a target distribution.
> Consequently, their associated vector fields and score functions are intractable. They need to be approximated by a neural network trained using ground-truth samples from the target distribution.
> Note that these methods assume access to ground-truth samples from the target distribution, while in our setting, we only have access to the unnormalized density. Therefore, standard diffusion methods are not applicable without relying on expensive adaptations.
>
> **Conditional Paths and Tractability.**
> A core novelty of our work is Conditional Interpolants, which follow a *conditional* probability path, instead of a marginal probability path.
> Because we condition on the reference sample $z$, the associated transport dynamics are tractable.
> This means that they are governed by a score function that admits exact, closed-form expressions derived directly from the target density.
> Therefore, CDS requires no neural networks, simulation-free training, or learning of any kind. The score is simply evaluated, not learned.
>
>
>
> **2. The role of Parallel Tempering (PT)**
>
> We emphasize that our framework is not restricted to PT, and any MCMC method can be used for initialization. We next explain why this step is needed and why annealing-based methods (such as PT) are particularly effective.
>
> **Why do we need it?**
> The conditional vector field driving the process is singular at $t=0$, causing trajectories to diverge if we integrate from $t=0$. This issue also arises in flow matching and diffusion models, where the loss becomes singular at the time boundaries.
> Therefore, we initialize the integration at a small $t_0 > 0$.
> Initializing at $x_{t_0} = z$ performs poorly, as this initial condition is inconsistent with the inner dynamics.
> Instead, we draw samples from the initialization distribution $\nu_{t_0 \mid z}$ using an MCMC method.
>
> **Why is it highly effective?**
> As $t_0 \to 0$, the initialization distribution $\nu_{t_0 \mid z}$ concentrates around $z$. Because of this, there is a strong overlap between the reference distribution and successive annealing distributions, making annealing-based methods especially effective.
> We demonstrate this for PT in Fig. 4 of the paper.
> Additionally, in the rebuttal, we have performed an additional experiment where we compare PT with other methods for initialization (see response to Rev. up6f).
>
> In summary, we use PT to "jump" this short distance, after which the SDE transports samples to the target.
>
> **Differences from Rissanen et al. (2025).**
> In "Progressive Tempering Sampler with Diffusion", the authors train a diffusion model starting from samples from a high-temperature distribution, gradually decreasing the temperature until reaching the target distribution.
> Their method relies on a "temperature guidance" mechanism to transform the samples. PT serves only as a baseline for comparison, not as part of their proposed method.
> Our proposed CDS is fundamentally different: we do not train a neural network, and we use PT solely for the highly efficient initialization step at $t_0$, relying entirely on our closed-form SDE to transport the samples.
>
> **3. Predictor-Corrector Steps**
>
> The intuition of the reviewer is correct. Stage 2 of our method is an Euler-Maruyama discretization of the closed-form SDE dynamics.
>
> **The advantage of our method.**
> Similar to standard diffusion models, we augment integration with MCMC corrector steps to mitigate discretization errors. A key advantage of our approach is that the exact (unnormalized) target density is available at each time step, allowing us to incorporate exact Metropolis-Hastings acceptance steps in the corrector (unlike the approximate correctors used in diffusion models).
>
> As our experiments in Appendix J.2 (Figure 12) show, this can lead to significant performance improvements in some tasks (ALDP).
> Furthermore, additional experiments (see response to Rev. up6f) show that corrector steps mitigate the effects of a suboptimal choice of the diffusion variance $\sigma_t^2$.
>
> ---
> We hope this response resolves the reviewer’s concerns and makes our contributions clearer. We are happy to expand on any point during the discussion period and kindly ask the reviewer to reconsider their score if satisfied.

---

> > ### Author Rebuttal · Reviewer_sovk · 2026-04-04
> >
> > Thank you to the authors for their thoughtful response. However, I am still a bit concerned about the novelty of this work and the presentation of the paper could be improved vastly to convey the idea more clearly. However, as I am a bit uncertain about this work, I will leave my score and confidence level as it is.

---

> > > ### Author Response · Authors · 2026-04-05
> > >
> > > Thank you for engaging with our paper! We appreciate the feedback and would like to clarify the remaining concerns.
> > >
> > > **1. Novelty**
> > >
> > > Our paper addresses sampling from complex probability distributions. Its novelty lies in a new transport construction for mapping samples from a reference distribution to the target distribution, together with a practical sampling pipeline built on top of it.
> > >
> > > **Training-free diffusions / flows.** We introduce Conditional Interpolants, a new class of stochastic processes with tractable transport dynamics that do not require training any neural network. This differs fundamentally from standard diffusion models and flow-matching methods, which assume access to ground truth samples to train an intractable score or vector field. Our construction yields a process that can be simulated directly, without learned parameters. In this sense, the key novelty is replacing learned transport with explicit, simulation-ready transport dynamics.
> > >
> > > **Combining diffusion-based transport with annealing-based sampling.** CDS combines two widely used paradigms for sampling from complex distributions: Parallel Tempering (PT) to sample from the initialization distribution (Stage 1), and the Conditional Interpolant SDE to transport these samples to the target distribution (Stage 2). This combination is novel, and it leverages the complementary strengths of both methods: PT improves exploration of complex landscapes, while the Conditional Interpolant SDE provides a principled transport mechanism. Together, they achieve state-of-the-art performance in terms of computational cost, measured by the number of density evaluations, versus sampling quality.
> > >
> > > In this rebuttal, we have also experimentally confirmed the validity and design of both stages. In our response to Rev. up6f, we discuss the importance of Stage 1 and why PT is particularly effective. In our response to Rev. DFyx, we explain why the Conditional Interpolant SDE is a principled transport mechanism and why it outperforms the alternative inverse-transform approach.
> > >
> > >
> > > **Theoretical grounding.** Our method is supported by a rigorous analysis, including proofs showing that the initialization transport cost vanishes at short diffusion times. This is an important part of the contribution, because it explains why the two-stage procedure is computationally viable and not just heuristic.
> > >
> > > **2. Presentation and clarity**
> > >
> > > **On the clarity of the paper.** We agree that clarity is essential, and we designed the paper to be accessible to a broad audience. For that reason, we included background and notation needed to follow the derivation in Section 2, and we structured the presentation to build intuition progressively: from the derivation of Conditional Interpolants (Section 3.1), to their transport dynamics (Section 3.2), their initialization (Section 3.3), and finally the practical algorithm in Algorithm 1 (Section 3.4). We also used Figures 1 and 3 to present the two-stage CDS procedure at a high level before introducing the full mathematical details. In addition, we use the linear interpolant as a running example throughout the paper, highlighted in blue boxes for ease of reference.
> > >
> > > **How we addressed clarity concerns.** We understand that, for a novel method that spans several technical areas, clarity is an ongoing challenge, and we have made a substantial effort to address this in the revised version. Based on feedback from other reviewers, the main sources of confusion were:
> > > 1) The distinction between the Conditional Interpolant SDE and the corresponding inverse transform, which is a natural alternative transport method. We address this in our response to Rev. up6f, where we explain the benefits of the Conditional Interpolant SDE and why it outperforms the inverse transform.
> > > 2) The need for a general formulation of the Conditional Interpolant framework. We address this in our response to Rev. DFyx, where we explain why a general formulation is necessary, both for better interpolants and for sampling in non-Euclidean spaces.
> > >
> > > Both reviewers indicated that our response resolved their concerns about presentation and clarity, suggesting that the revised version is clearer and more accessible. We will incorporate these clarifications into the camera-ready version to further strengthen the presentation and highlight the paper’s contributions. We would also be grateful for any additional specific suggestions to improve the final version.
> > >
> > > ---
> > >
> > > We hope this clarifies the remaining concerns. Given these clarifications and the improvements already made in the revised version, we would be grateful if you would consider updating your evaluation accordingly.

---

### Official Review · Reviewer_DFyx · 2026-03-08

**Soundness:** 2
**Presentation:** 3
**Significance:** 2
**Originality:** 3
**Overall Recommendation:** 4
**Confidence:** 4

**Summary:**

This paper explores sampling from unnormalized densities where the goal is generate high quality samples using as few expensive energy function evaluations as possible. The paper proposes a 2-stage scheme: initialize a distribution using Parallel Tempering (PT) for global exploration, then refine the samples using a diffusion / annealed langevin SDE dynamics. Both the PT and the refining or diffusion stage only require evaluating the energy function, and do not require neural training. The paper compares to other non-neural samplers and reports a better trade-off between sample quality and number of energy function evaluations.

**Compliance With Llm Reviewing Policy:**

Affirmed.

**Final Justification:**

I appreciate the new ablation, comparison methods, and clarification on hyper-parameter tuning. The approach seems to be interesting and nontrivial, and hopefully refining the presentation will further help readers appreciate the contribution.

**Key Questions For Authors:**

My key questions are:
- Have I mis-understood the method above? The conditional distribution is just scaled, so the "conditional interpolant" idea is not really fundamentally changing anything about the sampling problem.
- Can you clarify on the choice of methods for comparison, and the hyper-parameter tuning done for comparisons, as discussed above?

**Limitations:**

yes

**Strengths And Weaknesses:**

# Strengths:
- Combining parallel tempering and diffusion was interesting to me.
- The framing of the problem and the development of the method was easy to follow.
- I'm not very familiar with this literature but found the related work discussion useful


# Weaknesses:

## Presentation and significance of the method
The presentation of the method seems to unnecessarily obfuscate what is happening. The final approach doesn't seem to require conditional interpolants or diffusion at all. Please let me know if my understanding below is wrong.

1. Use a random z to shift and scale the distribution. Use parallel tempering on this scaled distribution. (Global exploration)
2. Refine the sample further by using an annealed Langevin sampler with a MH correction. The SDE also scales the distribution back to the data scale. (Local refinement)

The conditional interpolant framing obscures the fact that the distribution at different times t is not really changed at all, it is a scaled version of the original distribution. Fig. 1 shows that the first step samples the scaled distribution, and the second step mostly just scales it back, but with some local exploration refining the sample.

Interpretations under my explanation:
- Scaling the data shouldn't have much effect on the Parallel Tempering (PT). Changing "t0" only acts as a scaling preconditioner. I interpret Fig. 4 as testing the preconditioner. The distribution is not changing, only the scale. Please let me know if I misunderstood this.
- A sample from the PT stage should *directly* be a sample from the target distribution, through the trivial linear transform.
- Instead of using the linear transform, you use an SDE to transport to the unscaled target distribution. This has the effect of providing some additional Langevin like local sampling to refine the sample.
- In principle, the scaling (conditional interpolant) could be completely removed from the framing. We just do PT with different pre-conditioning, then a MALA like refiner, also possibly with preconditioning.

I thought there should be papers that already do PT + Langevin refinement. But at first glance, I could only find papers that interleave PT and Langevin. So from that point of view, the method presented is still somewhat novel.

Despite the fact that it can be understand more directly, I still support including the conditional interpolant framing, as it points to an interesting way to use different interpolant functions. Though I'm skeptical that more complex functions with tractable Jacobians would be useful or easy to find, and the paper doesn't give any examples. That would increase the significance of the paper, I think.


## Comparisons
I am not very familiar with this field but was a little disappointed with the comparisons.
- HMC / MALA seem like weak baselines. One could include refinements like NUTS, for example.
I think the "Microcanonical HMC" and variants also look promising, though those are quite new and maybe excel more at local refinement rather than global exploration (https://arxiv.org/pdf/2503.01707 e.g.).
- Would Stein methods be an appropriate comparison?
- Methods that combine PT and langevin would make sense. It seemed like there were many papers when I searched, but maybe they are inappropriate for some reason (an example https://proceedings.mlr.press/v119/deng20b/deng20b.pdf , refers to "replica exchange MC" as parallel tempering)
- The idea to exclude neural samplers was based on the Rissanen paper. At first glance this seems a reasonable restriction, as it is hard to compare these methods directly. (It seems neural samplers have to generate training samples, but then after training the generation cost is amortized. This makes it hard to compare directly, I agree.)

Although the presented results are somewhat better, I suspect this is a case of more hyper-parameter tuning on the proposed method, compared to the competitors. Table 3 is hard to understand - it says "hyperparameters considered", so I tend to think this means the hyperparameter search grid. Most lines have only a single value, but the value is different across datasets. So I can't tell how big the grid was, or how the different parameters were determined for each dataset. Notably, CDS has the most hyper-parameters, AND it inherits all the hyperparameters from NRPT. MALA / HMC use one or zero hyper-parameters. As per my point above, I believe more modern variants could actually excel, especially in the high-d BNN case.


## Other comments
- The title gave me a completely wrong idea about what this paper is about.

---

> ### Author Rebuttal · Authors · 2026-03-30
>
> We thank the reviewer for the feedback. We address the concerns below.
>
> **Transparency of the formulation.** Our goal was to balance rigor and accessibility. The linear interpolant acts as a running example, highlighted in blue boxes for clarity. Thus, we did not intend to obscure its "shift and scale" effect; in Section 3.3 (lines 269-272) we explicitly state: "(the intermediate distribution) retains the multimodal structure of $\nu$, differing only by a global contraction."
>
> **Why a general formulation?** Conditional Interpolants are designed for broad applicability. This generality matters in at least two ways:
> 1) To design **better interpolants**, such as geometrically informed ones (e.g., respecting periodic boundaries in molecular dynamics).
> 2) Sampling **in non-Euclidean spaces**, such as manifolds, spaces with hard constraints, or discrete spaces. In such settings, interpolants are not as straightforward as in the Euclidean case (e.g., on manifolds they can be defined via geodesics, with dynamics governed by the metric). We will clarify this in the paper.
>
> **Using the inverse transform.** The reviewer suggests a related baseline: directly inverting the linear transform after Stage 1. However, a deterministic map preserves or even amplifies PT mixing errors, while the SDE refines and mixes samples along the trajectory.
> This is confirmed experimentally in the response to Rev. Cc9V, where CDS's SDE integration performs substantially better. We will include these results in the final version.
>
> **Removing the "scaling".** The reviewer suggests using preconditioned PT on the target followed by a preconditioned MALA-like sampler (PT+MALA). The difference with CDS (PT+SDE) is that CDS runs PT on a distribution closer to the reference, which improves efficiency and mode exploration.
>
> The table below shows the main metrics of both approaches under a fixed density-evaluation budget for each task: $W_2$ for GM and LJ, KL Ram. for ALDP, and test NLL for BNN (lower is better). The Pareto fronts, analogous to Fig. 5 in the paper, are [here](https://anonymous.4open.science/r/re-3381/5.png). They confirm that CDS consistently outperforms PT + MALA, and will be included in the paper.
> ||PT+SDE|PT+MALA|
> |-|-|-|
> |GM-16|7.08±0.53|7.74±0.50|
> |LJ-55|1.78±0.01|1.80±0.01|
> |ALDP|0.09±0.03|0.11±0.00|
> |BNN|0.53±0.07|0.56±0.42|
>
> **Baselines.** We selected state-of-the-art baselines (NRPT, OASMC, DiGS) and standard MCMC methods (MALA, HMC) as a performance floor, since we expected them to struggle on multimodal tasks.
>
> We added the suggested baselines: NUTS, MAMS (Microcanonical HMC), and SVGD (Stein Variational Gradient Descent). We omitted methods that combine approximate Langevin dynamics with PT, such as the suggested "Non-convex Learning via Replica Exchange Stochastic Gradient MCMC," because they address a different setting in which the target density is available only through stochastic gradients.
>
> The table below reports the main performance metrics under a fixed density evaluations budget for each task (Pareto fronts are [here](https://anonymous.4open.science/r/re-3381/6.png)). CDS outperforms or is competitive with all baselines, offering the best quality–cost trade-off. We will include these results in the paper.
> ||CDS (ours)|NUTS|MAMS|SVGD|
> |-|-|-|-|-|
> |GM-16|6.99±0.5|104.1±0.06|109.0±0.15|110.9±0.07|
> |BNN|0.38±0.23|0.37±0.29|0.44±0.23|0.97±0.40|
>
> **Hyperparameter tuning.** We thank the reviewer for raising this. We clarify that CDS did not benefit from an unfair hyperparameter search:
> - **Grid search.**
> We apologize for the confusion.
> Table 3 lists configurations selected via grid search, rather than the full grid (which is available [here](https://anonymous.4open.science/r/re-3381/7.png)). These recommended settings ensure reproducibility without running the full grid. We will update it in the paper for transparency.
> - Consequently, hyperparameters **vary across datasets**, not only for CDS, but also for other methods. Importantly, $\sigma_t$ varies since it was fixed to the minimum stable value per task (see response to Rev. up6f).
> - **Number of hyperparameters.** For a fair comparison, CDS's PT hyperparameters were fixed to those optimized for NRPT (see the table caption). They are likely suboptimal, since CDS targets a distribution closer to the reference, possibly requiring fewer replicas. In the recommended settings, CDS uses fewer configurations per budget (4) than NRPT and OASMC (8).
> - **MALA / HMC, modern variants:** poor performance is not due to suboptimal hyperparameters, but rather to their mode-trapping behavior; see the Baselines point.
>
> **Title.** It highlights that CDS dynamics follow a conditional probability path. A more descriptive title would be: "CDS: Sampling Unnormalized Densities via Conditional Interpolants." We are open to suggestions.
>
> ---
> We hope this response clarifies the concerns. We are open to further discussion. If satisfied, we kindly ask the reviewer to reconsider the score.

---

> > ### Author Rebuttal · Reviewer_DFyx · 2026-04-02
> >
> > Thanks for the detailed follow-up. I like the ablation, so we can see that we gain something with the PT + SDE combo, over PT + scaling. I appreciate the extra comparison methods, and the clarifications on the hyper-parameter tuning.  I will raise my score. I see that some of my confusions from the presentation were shared by other reviewers, I hope this can be useful in refining the final version.

---

> > > ### Author Response · Authors · 2026-04-05
> > >
> > > Thank you for your comments! We are glad that our rebuttal addressed your concerns.
> > >
> > > Together with feedback from the other reviewers, your suggestions have been very helpful in improving the paper, particularly in the presentation of our method. We have clarified the need for a general formulation and better explained how our approach provides a principled framework for extending Conditional Interpolants beyond the linear/Euclidean setting. We have also substantially improved the explanation of the second stage of CDS. While this was partially discussed in Appendix J of the original version, this suggestion motivated a clearer theoretical treatment and additional ablations, which significantly strengthen this component. We sincerely appreciate this feedback, as it has led to a clearer presentation of both the method and its contributions.
> > >
> > > Overall, we believe these revisions significantly improve the clarity and technical strength of the paper. We would greatly appreciate your support in your final assessment.
> > >
> > > Thank you again for your time and constructive feedback!

---

### Official Review · Reviewer_up6f · 2026-03-10

**Soundness:** 4
**Presentation:** 3
**Significance:** 3
**Originality:** 3
**Overall Recommendation:** 4
**Confidence:** 3

**Summary:**

This paper introduces Conditional Diffusion Sampling (CDS), a novel algorithm designed to sample from unnormalized Boltzmann distribution. Traditional annealing-based methods  suffer from expensive simulation when reference distribution and target distribution has high discrepancy. CDS leverages parallel tempering and stochastic interpolants and shows efficient robust results without  neural network training. Empirical evaluations on diverse tasks and distributions showsthat CDS improves sampling efficiency.

**Compliance With Llm Reviewing Policy:**

Affirmed.

**Final Justification:**

I thank the authors for the detailed response and the additional clarifications on the numerical results. The rebuttal addresses most of my concerns and strengthens the experiment section.

**Key Questions For Authors:**

* Eq.(5): Not a question, but wouldn't the _stochastic_ interpolant requires additional noise? It seems interpolation between to sampled points without any extra stochasticity.
* Figre 5: the performance gain of CDS on BNN is much larger than on the other tasks. Does CDS perform better in high-dimensional setting?

**Limitations:**

yes

**Strengths And Weaknesses:**

**Strengths**
* Overall, the paper is clearly written and technically sound.
* The proposed method presents a thoughtful and well-justified approach to Boltzmann distribution sampling by combining traditional sampling method with recent advances in generative modeling.
* The empirical results presented in the paper are convincing.

**Weaknesses**

*  Compared to the Stage 2 SDE integration part, the conditional sampling part is explained less clearly. The authors argue that MCMC methods mix poorly. However, as I understand, the conditional sampling can be performed using any annealing-based methods. Is the choice of conditional sampling strategy is crucial to the overall performance?
* As it stands, Algorithm 1 appears to perform additional corrector steps, which might incurs additional energy evaluations. Is this necessary for the strong performance? Compared to neural samplers, how many energy evaluation does CDS require per sampled particle? I understand the advantage of being able to sample without neural network training, but it would be helpful to better quantify the trade-off against neural samplers in terms of sampling performance and the number of energy evaluations required at inference.
* I believe experimental section would benefit from more quantitative results. In particular, it would be useful to evaluate the resulting evaluation metric (e.g., 2-Wasserstein distance) for the different values of $t_0$. While, the authors provided round trip count, it would be much intuitive for the reader how the choice of $t_0$ affect the distance metric.

---

> ### Author Rebuttal · Authors · 2026-03-30
>
> We appreciate the feedback and thank the reviewer for it. Below we address their concerns.
>
> **Conditional Sampling (Stage 1).**
> As the reviewer points out, our approach is not limited to PT in Stage 1, and any sampling strategy could be used.
> Since $\nu_{t_0 \mid z}$ remains multimodal, standard MCMC methods (such as MALA or HMC) are expected to mix poorly. Instead, annealing-based methods (such as PT and SMC) that bridge the reference and the target distributions are better suited, since $\nu_{t_0 \mid z}$ approaches the reference as $t_0\to 0$.
>
> In our experiments, PT was particularly effective, but we had not included this study in the paper; we will update Appendix J accordingly.
> We ran an ablation on the GM task comparing PT, SMC, and HMC.
> The table reports top-performance results (budget=$10^5$ density evaluations, $W_2$ as the metric, lower is better) in GM-2 and GM-16 (Pareto fronts [**here**](https://anonymous.4open.science/r/re-3381/2.png)):
> ||CDS w/ PT|CDS w/ SMC|CDS w/ HMC|
> |-|-|-|-|
> |GM-2|0.67±0.01|1.7±0.4|7.4±0.3|
> |GMNU-2|0.52±0.03|1.7±0.1|5.7±0.3|
> |GM-16|6.99±0.5|7.78±0.4|14.8±0.3|
> |GMNU-16|6.09±0.22|26.9±0.8|68.4±0.6|
>
> These results suggest that PT provides better initialization than SMC and HMC, with HMC often getting trapped in local modes.
>
> **Corrector Steps.**
> Corrector steps reduce discretization bias in the Euler–Maruyama approximation and help intermediate samples match $\pi_{t \mid z}$. As shown in Appendix J.2 (Figure 12), one corrector step has a negligible effect on GM and LJ, but improves performance on ALDP, likely due to higher sensitivity to integration errors. Their benefit also depends on the diffusion variance (smaller variance reduces the score's influence, making correctors more useful). We confirm this by varying the diffusion variance (see Pareto fronts [**here**](https://anonymous.4open.science/r/re-3381/3.png)). In the paper, we fixed $\sigma_t^2$ to the minimum stable value per task, so it was not part of the hyperparameter search. We will include this analysis in the final version.
>
> **Density Evaluations.**
> Total evaluations per particle is the sum of Stage 1 and Stage 2 costs. Because we cache states to avoid redundancy (all samplers use this), Stage 2 requires exactly $N(1+c)$ score evaluations ($N$=integration steps, $c$=corrector steps): 1 score evaluation for the SDE drift, plus 1 score and 1 density evaluation per MALA step. We will clarify this in the paper.
>
> **Comparison with Neural Samplers.**
> Neural samplers amortize sampling by training a neural network using both samples (from the model or previous iterations) and the target density. As we argue in Sec. 4.2, this training is computationally expensive. Once finished, they can generate samples with a single forward pass and no additional density evaluations.
>
> Our method is orthogonal and complementary to them, as CDS samples could be used to initialize the training.
> As the two approaches operate under different regimes, we chose not to include a direct comparison. We will include this discussion in the final version.
>
> **Quantitative Results.**
> In this rebuttal we have substantially expanded the experimental study.
> Alongside the experiments described above, we added three new baselines (NUTS, MAMS, SVGD; see Rev. DFyx) and deepened the Stage 2 analysis (see Rev. CC9V and Rev. DFyx).
> Also, we have conducted the experiment suggested, evaluating the sample quality for different values of $t_0$. The table below reports the main performance metrics (lower is better), see the extended plot [**here**](https://anonymous.4open.science/r/re-3381/4.png).
> |$t_0$|GM-16 ($W_2$)|LJ-13 ($W_2$)|ALDP (Ram. KL)|BNN (Test NLL)|
> |-|-|-|-|-|
> |$1.0$|114.0|5.83|0.23|0.99|
> |$10^{-1}$|113.3|5.64|0.21|0.91|
> |$10^{-2}$|90.6|5.44|0.24|0.94|
> |$10^{-3}$|9.7|7.0|0.25|0.94|
> |$10^{-4}$|14.8|12.2|0.3|0.96|
>
> Our results match the paper’s PT efficiency analysis: decreasing $t_0$ improves performance due to the reduced transport, but too small values degrade it.
>
> **Other questions.**
> - **Stochastic Interpolants (SI) formulation.**
> Yes, the general SI framework includes an additional noise variable. Our formulation corresponds to setting the corresponding weight to zero. We thank the reviewer for raising this; we will clarify it in the paper.
> - **Performance on BNN.**
> For BNN, the prior serves as the reference distribution. Intuitively, the interpolant contracts the posterior toward the prior, reducing transport distance and facilitating mode exploration by PT.
> - **Dimensionality.** Regarding dimensionality, we expect CDS to inherit the same behavior as the sampler in Stage 1. For PT, inefficiency scales as $O(D^{1/2})$, as shown by Syed et al. (2021). We will add an experiment on the GM task to clarify this point.
>
> ---
> We hope the reviewer finds our response satisfactory. We are happy to elaborate further during the discussion period. If concerns are resolved, we kindly ask the reviewer to reconsider their score.

---

> > ### Author Rebuttal · Reviewer_up6f · 2026-04-03
> >
> > Thanks to the authors for the detailed response and the additional clarifications on the numerical results. The responses address most of my concerns, and I think they will strengthen the empirical section. Yet, I should note that while I am familiar with diffusion samplers, I am not familiar with some of the related references raised by the other reviewers.

---

> > > ### Author Response · Authors · 2026-04-05
> > >
> > > Thank you for your comments! We are happy to hear that your concerns have been adequately addressed, and that as a result, our paper now has a stronger empirical section.
> > >
> > > **About related references.** Regarding the related work raised by other reviewers, we would like to provide some brief context. Only Rev. DFyx suggested additional methods for comparison, which have now been included in the revised version. The suggested methods belong to the same class of samplers represented by MALA and HMC in our paper. These are local samplers, which typically struggle with mode-collapse and poor mixing in complex landscapes. Indeed, two of the suggested methods are variations of HMC, and the other can be interpreted as a variation of MALA. All of them have been integrated into the revised manuscript, and the new results confirm that our proposed CDS consistently outperforms them.
> > >
> > > Given that your concerns have been resolved, we would be grateful if you could consider updating your score to reflect the strengthened empirical results and, if you find the paper now compelling, supporting it in your final assessment. We believe the revised version presents a clearer and stronger case for CDS in terms of both performance and practical relevance.
> > >
> > > Thank you again for your time and constructive feedback.

---

### Official Review · Reviewer_Cc9V · 2026-03-10

**Soundness:** 2
**Presentation:** 1
**Significance:** 3
**Originality:** 3
**Overall Recommendation:** 4
**Confidence:** 1

**Summary:**

The authors propose a new sampling algorithm, Conditional Diffusion Sampling. This algorithm is composed of two different steps: first, a Parallel Tempering step to initialize an intermediate distribution followed by a Conditional Interpolants sampling step, which is similar to SDE generation methods but utilizes closed form steps instead of a neural network approximation. The authors demonstrate the capabilities of their method across several distributions and compare them to alternative sampling methods.

**Compliance With Llm Reviewing Policy:**

Affirmed.

**Final Justification:**

I thank the reviewers for their responses.
Ultimately, I find this paper to be a borderline case outside my area of expertise, and therefore maintain my low confidence. I slightly raise my score accordingly as I do not object to this paper's acceptance, and recommend following other reviewer's opinion. At the same time, I am not convinced that a revised version will fully alleviate my concerns on the paper's presentation and accessibility, as well as the positioning for the paper's contributions.

**Key Questions For Authors:**

See "Strengths And Weaknesses"

**Limitations:**

yes

**Strengths And Weaknesses:**

Strengths:
 * Based on the evaluations it seems that the proposed approach offers an advantage in sampling.
 * To the best of my knowledge, the proposed algorithm is novel.

Weaknesses:
 * The writing is hard to follow, making the motivation and novelty of the proposed approach unclear. The emphasis on a comparison with diffusion models gives the wrong impression, while the use of training-free is misleading, as training is usually used as a substitute for a closed form energy term used in sampling. Also, much of the relevant background is in the appendix and not the main paper.
 * What is the advantage of this method over applying Parallel Tempering on a shifted version of the reference or target distribution? This has been hard for me to understand from the paper

---

> ### Author Rebuttal · Authors · 2026-03-30
>
> We thank Rev. Cc9V for their suggestions. Next we address their concerns.
>
> **A General Clarification.** Our motivation comes from the high computational cost of sampling complex, unnormalized distributions. Annealing-based methods like Parallel Tempering (PT) explore well but struggle when reference and target distributions have little overlap (non-continuous transport). Conversely,  recent diffusion samplers offer continuous transport but their dynamics need to be approximated by training a neural network, which is costly in terms of density evaluations.
>
> Our work proposes combining the best of both worlds. The theoretical novelty lies in the derivation of Conditional Interpolants (CI), a new class of stochastic processes that admits an exact, closed-form solution for the transport dynamics, without requiring any neural network training.
> The proposed CDS leverages the global exploration of PT to initialize this process, after which samples are transported using the derived dynamics.
>
> **Relation to Diffusion Models.** We thank the reviewer for this point. We discuss diffusion models because our approach is closely related: both use a diffusion process to transport samples from a reference distribution to a target distribution. The key difference is that standard diffusion models assume access to target samples, which are used to train a score network. In our setting, we do not have samples, only the unnormalized density.
>
> Also, we discuss Neural Diffusion Samplers, which adapt this ideas to amortized sampling by training a neural network through an iterative optimization procedure involving both samples and the target density.
> Our method is orthogonal to these approaches and could be combined with them: CDS samples could be used as high-quality data to initialize the training of a neural sampler.
>
> **Why "training-free"?** By "training-free," we mean that CDS does not require any neural network training. Its dynamics are determined by the proposed CI, so the SDE drift is available in closed form and depends directly on the target score, with no training overhead.
>
> **Background in the Appendix.** We appreciate the feedback regarding the structure of the paper.
> In the main text (Section 2), we cover the core background necessary to understand our contribution: Markov Chain Monte Carlo (including Parallel Tempering), Diffusions, and Stochastic Interpolants.
> The proposed CDS is derived from these principles.
>
> Appendix A contains a revised description of the PT implementation. This material is useful for reproducibility but not essential to the main contribution. We will add a sentence pointing readers to Appendix A for the PT mechanics, and we are happy to move any specific material to the main text if the reviewer feels it is needed for clarity.
>
> **Advantages Over a Shifted Parallel Tempering Baseline.**
> We thank the reviewer for this baseline suggestion: applying PT to sample from the initialization distribution $\pi_{t_0 \mid z}$, and then simply map the samples back using the deterministic inverse transform of the interpolant ($F_{t_0 \mid z}^{-1}$).
> In our framework, this corresponds to replacing Stage 2 (the SDE integration) with a deterministic map.
>
> The advantage of CDS (PT + SDE) is that it refines the samples during transport. With the deterministic inverse transform, any imperfect mixing from PT is preserved, and may even be amplified. By contrast, SDE integration evaluates the target score along the trajectory, correcting mixing errors as samples are transported.
>
> We initially omitted this baseline because it is closely related to the steps-allocation ablation in Appendix J.2. However, we agree it should be evaluated explicitly, and we have now run the exact ablation suggested by the reviewer. We will include the results in the final version.
>
> The table below reports the main performance metrics ($W_2$ for GM and LJ, KL Ram. for ALDP, and test NLL for BNN; lower is better for all metrics) under a fixed density evaluations budget for each task (Pareto fronts, analogous to Fig. 5 in the paper, are [**here**](https://anonymous.4open.science/r/re-3381/1.png)).
>
> ||PT+SDE|PT+Inv.Transform|
> |:---|:---|:---|
> |GM-2|0.67±0.01|0.71±0.11|
> |GMNU-2|0.54±0.02|0.60±0.03|
> |GM-16|7.08±0.53|7.18±0.69|
> |GMNU-16|6.12±0.3|6.45±0.64|
> |LJ-13|0.79±0.01|0.79±0.01|
> |LJ-55|1.78±0.01|1.84±0.02|
> |ALDP|0.09±0.03|0.16±0.01|
> |BNN|0.53±0.07|0.8±0.26|
>
> While PT + Inv. Transform provides a reasonable baseline, the refinement of the SDE in CDS yields consistently superior performance.
>
> ---
> We will include these discussions and results in the final version. We hope this addresses the reviewer’s concerns and clarifies our contributions, and we are happy to elaborate further during the discussion period. If concerns are resolved, we kindly ask the reviewer to reconsider their score.

---

> > ### Author Rebuttal · Reviewer_Cc9V · 2026-03-31
> >
> > Based on the other reviews, it seems that the paper's proposition was not well stated in the manuscript, leading to some confusion. I believe the paper would be much stronger with an improved presentation. I also believe the authors have misunderstood my comment on applying Parallel Tempering on a shifted version, as also raised by Reviewer DFyx.

---

> > > ### Author Response · Authors · 2026-04-02
> > >
> > > We thank you for the follow-up and for taking the time to engage with our rebuttal. We appreciate your perspective, as it helps us further improve the work.
> > >
> > > We address below the remaining concerns raised in the follow-up, expanding on our rebuttal.
> > >
> > > **Applying PT on a shifted version.** We apologize for any confusion in our previous explanation. When using the linear interpolant, the intermediate distributions become a shifted and scaled version of the original distribution. Thus, in the experiment we performed in the rebuttal, we are indeed applying PT on a shifted version of the original distribution. Once we have obtained samples from that distribution, we need to transform the samples back to the original space. This can be done in two ways:
> > > 1) Applying **the inverse shift and scale transformation** directly to the samples from the intermediate distribution, which is what we did in the experiment. This corresponds to the **PT + Inv. Transform** column in the table.
> > > 2) Simulating the interpolant **SDE dynamics** with those samples as initial conditions. This corresponds to the **PT + SDE** column in the table
> > >
> > > In the case you had another approach in mind for "applying PT on a shifted version", we would welcome clarification on the specific approach you had in mind, and we would be willing to incorporate any variation that you suggest.
> > >
> > > Also, we would like to expand on the theoretical reasons on why Conditional Diffusion Sampling is significantly more effective than applying PT on a shifted version of the original distribution.
> > >
> > > **The deterministic transformation.** Conceptually, applying the inverse transformation to the samples obtained from the intermediate distribution is perfectly valid. If those samples were perfect, applying the inverse transformation would yield perfect samples from the original distribution. In practice, however, the samples will be far from perfect, and the inverse deterministic transformation will amplify these errors.
> > >
> > > **The value of SDE dynamics.** As we noted in the rebuttal, the SDE dynamics offer a fundamental advantage: they continuously correct the samples as they evolve, improving mixing and convergence to the target distribution. This is one of the main reasons why diffusion and flow-matching models work well in practice and is consistent with the general framework of stochastic interpolants by Albergo et al. (2025). Our method builds on this principle.
> > >
> > > Moreover, the SDE stage introduces flexibility through the $\sigma_t$ parameter, which controls stochasticity and helps avoid mode collapse inherited from the PT stage, unlike a deterministic transformation, which cannot recover from such collapse. Discretization errors are also mitigated by our predictor-corrector scheme, which makes the method robust to poor choices of $\sigma_t$. This is evidenced by our ablation study in Appendix J.2 (Figure 12), where we show that corrector steps significantly improve performance for ALDP, likely due to higher sensitivity to integration errors. We discuss this further in our response to Rev. up6f.
> > >
> > > We thank you for raising this discussion, as it helped us improve the clarity of the SDE stage. We will incorporate this clarification in the final version of the paper.
> > >
> > > **A note on presentation and clarity.** We acknowledge that making the method accessible to readers with varied backgrounds is an ongoing challenge, as reflected in the reviewers' feedback on the presentation dimension. We welcome any specific suggestions on where the presentation could be improved for the camera-ready version.
> > >
> > > We hope this response clarifies the raised concerns.

---

### Decision · Program_Chairs · 2026-04-30

**Decision:**

Accept (regular)

**Comment:**

Most of the reviewers concerns were related to the numerical experiments and to the presentation of the paper. In particular, some concerns were raised about the lack of baselines (improved MCMC) and also by the fact that performance could be related to hyperparameter tuning more than a new methodological aspect. Reviewers were also partly concerned by the presentation and motivation of the contribution and its novelty with existing works.

During rebuttal, the authors provided additional experiments, which substantially expanded the experimental study. They added state-of-the-art MCMC appraoches (NUTS, Microcanonical HMC), and SVGD (Stein Variational Gradient Descent) which confirm the claims provided in the original paper. The authors also clarified the hyperparameter tuning for all methods, and provided additional details on the performance of the MCMC methods which are known to be stuck in local modes in challenging settings, even with hyperparameter tuning. The authors also clarified the position of their work highlighting how their approach combines  well-grounded  procedures to sample from complex distributions (Parallel Tempering and the Conditional Interpolant SDE). The proposed method is supported by a rigorous analysis and reaches state-of-the art performance.
I believe that this paper provides a valuable contribution to the Machine Learning community, and encourage the authors to clarify their experimental section and position of their work to support the originality of their procedure.